# Greenland ice sheet mass balance from 1840 through next week

Kenneth D. Mankoff[1], Xavier Fettweis[2], Peter L. Langen[3], Martin Stendel[4], Kristian K. Kjeldsen[1], Nanna B. Karlsson[1], Brice Noël[5], Michiel R. van den Broeke[5], Anne Solgaard[1], William Colgan[1], Jason E. Box[1], Sebastian B. Simonsen[6], Michalea D. King[7], Andreas P. Ahlstrøm[1], Signe Bech Andersen[1], and Robert S. Fausto[1]

[1]Department of Glaciology and Climate, Geological Survey of Denmark and Greenland (GEUS), Copenhagen, Denmark
[2]SPHERES research unit, Department of Geography, University of Liège, Liège, Belgium
[3]Department of Environmental Science, iClimate, Aarhus University, Roskilde, Denmark
[4]Danish Meteorological Institute (DMI), Copenhagen, Denmark
[5]Institute for Marine and Atmospheric Research, Utrecht University, The Netherlands
[6]Geodesy and Earth Observation, DTU Space, Technical University of Denmark, Lyngby, Denmark
[7]Polar Science Center, University of Washington, Seattle, WA, United States

**Correspondence:** Ken Mankoff (kdm@geus.dk)

**Abstract.** The mass of the Greenland ice sheet is declining as mass gain from snow accumulation is exceeded by mass loss from surface meltwater runoff, marine-terminating glacier calving and submarine melting, and basal melting. Here we use the input/output (IO) method to estimate mass change from 1840 through next week. Surface mass balance (SMB) gains and losses come from a semi-empirical SMB model from 1840 through 1985, and three regional climate models (RCMs; HIRHAM/HARMONIE, MAR, and RACMO) from 1986 through next week. Additional non-SMB losses come from a marine terminating glacier ice discharge product and a basal mass balance model. From these products we provide an annual estimate of Greenland ice sheet mass balance from 1840 through 1985 and a daily estimate at sector and region scale from 1986 through next week. This product updates daily and is the first IO product to include the basal mass balance which is a source of an additional ~24 Gt yr$^{-1}$ of mass loss. Our results demonstrate an accelerating ice-sheet-scale mass loss and general agreement (coefficient of determination, r$^2$, ranges from 0.62 to 0.94) among six other products, including gravitational, volume, and other IO mass balance estimates. Results from this study are available at https://doi.org/10.22008/FK2/OHI23Z (Mankoff et al., 2021).

## 1 Introduction

Over the past several decades, mass loss from the Greenland ice sheet has increased (Khan et al., 2015; The IMBIE Team, 2019). Different processes dominate the regional mass loss of the ice sheet, and their relative contribution has fluctuated in time (Mouginot and Rignot, 2019). For example, in the 1970s nearly all sectors gained mass due to positive SMB, except the northwest sector where discharge losses dominated. More recently in the 2010s, all sectors lost mass, with some sectors losing mass almost entirely via negative SMB, and others primarily due to discharge (Fig. 1).

There are three common methods for estimating mass balance – changes in gravity (Barletta et al., 2013; Groh et al., 2019; The IMBIE Team, 2019; Velicogna et al., 2020), changes in volume (Simonsen et al., 2021a; Sørensen et al., 2011; Zwally and

Giovinetto, 2011; Sasgen et al., 2012; Smith et al., 2020), and the input/output (IO) method (Colgan et al., 2019; Mouginot et al., 2019; Rignot et al., 2019; King et al., 2020). Each provides some estimate of where, when, and how the mass is lost or gained, and each method has some limitations. The gravity mass balance (GMB) estimate has low ~100 km spatial resolution (where), monthly temporal resolution (when), and little information on the processes contributing to changes in mass balance components (how). The volume change (VC) mass balance estimate has ~1 km spatial resolution (where), often provided on annual or multi-year temporal resolution (when), and little information on the driving processes (how).

The IO method has a complex spatial resolution (where). The inputs typically come from regional climate models (RCMs) which can reach a spatial resolution of up to 1 km. However, that spatial resolution is generally reduced in the final output to sector or region scale – typically higher than GMB but now lower resolution than VC. The IO temporal resolution (when) is limited by ice velocity data updates, which for the past several years occur every 12 days year-round after the launch of the Sentinel missions (Solgaard et al., 2021). The primary issue with the IO method is unknown ice thickness in some locations (e.g. Mankoff et al. (2020b)). Finally, the IO method can provide insight into the processes (how) by distinguishing between changes caused by SMB (which may be due to changes in positive and/or negative SMB components) vs. changes in other mass loss terms (e.g., calving). Our IO method is also the first IO product to include the basal mass balance (Karlsson et al., 2021) – a term implicitly included in the GMB and VC methods but neglected by all previous IO estimates.

In this work we introduce the new PROMICE Greenland ice sheet mass balance dataset based on the IO method, updating the previous product from Colgan et al. (2019). We use the SMB field from one empirical model from 1840 through 1985, and three RCMs from 1986 onward. The combined SMB field used here is comprised of positive SMB terms (precipitation in the form of snowfall, rainfall, condensation/riming, and snow drift deposition) and negative SMB terms (surface melt, evaporation, sublimation, and snow drift erosion). We also use the basal mass balance, and an estimate of dynamic ice discharge. Spatial resolution is effectively per sector (Zwally et al., 2012) or region (Mouginot and Rignot, 2019). Temporal resolution is annual from 1840 through 1985, and effectively daily since 1986 - the RCM fields are updated daily and forecasted through next week, and the discharge at marine terminating glaciers is updated every 12 days with ~12 day resolution, interpolated to daily, and forecasted using historical and seasonal trends through next week. Thus, this study provides a daily-updating estimate of Greenland mass changes from 1840 through next week.

## 2 Terminology

We use the following terminology throughout the document:

- This Study refers to the new results presented in this study.

- Recent refers to the new 1986 through next week daily temporal resolution data at region and sector scale

- Reconstructed refers to the adjusted Kjeldsen et al. (2015) annual temporal resolution data at ice sheet scale used to extend this product from 1986 back through 1840. The 1986 through 2012 portion of the Kjeldsen et al. (2015) data set is used only to adjust the reconstructed data, then discarded.

- ROI (region of interest) refers to one or more of the ice sheet sectors or regions (Fig. 1).

- Sector refers to one of the Zwally et al. (2012) sectors (Fig. 1), expanded here to cover the RCM ice domains which exist slightly outside these sectors in some locations.

- Region refers to the Mouginot and Rignot (2019) regions (Fig. 1), expanded here to cover the RCM ice domains.

- SMB is the surface mass balance from an RCM, or the average of multiple RCM SMBs. The use should be clear from the context.

- D is solid ice discharge. It includes both calving and submarine melting at marine terminating glaciers.

- BMB is the basal mass balance. It comes from geothermal flux ($BMB_{GF}$), frictional heating from ice velocity ($BMB_{friction}$), and viscous heat dissipation ($BMB_{VHD}$).

- MB is the total mass balance including the BMB term (Eq. 3).

- MB* is the mass balance not including the BMB term (Eq. 4).

- HIRHAM/HARMONIE, MAR, and RACMO refer to the RCMs, which only provide SMB, and runoff in the case of MAR. However, when referencing the different MB products, we use, for example, "MAR MB" rather than repeatedly explicitly stating "MB derived from MAR SMB minus BMB and D". The use should be clear from the context.

## 3 Product description

The output of this work is two NetCDF files and one CSV file containing a time series of mass balance and the components used to calculate mass balance. The only difference between the two NetCDF files is the ROI – one for Zwally et al. (2012) sectors and one for Mouginot and Rignot (2019) regions. Each NetCDF file includes the ice sheet mass balance (MB), MB per region of interest (ROI; sector or region), MB per ROI per RCM, ice sheet surface mass balance (SMB), SMB per ROI, ice sheet discharge (D), D per ROI, ice sheet basal mass balance (BMB), and BMB per ROI. The CSV file contains a copy of the ice-sheet summed data.

An example of the output is shown in Fig. 2, where the top panel shows mass balance for the entire Greenland ice sheet, in addition to SMB, and D, at annual resolution. The lower panel shows an example two years at daily temporal resolution. The ice-sheet-wide product includes data from 1840 through next week, but the sector and region-scale products only includes data from 1986 through next week, because the 1840 through 1985 reconstructed only exists at ice-sheet scale (Fig. 1).

## 4 Data sources

This section introduces data products that exist prior to and are external to this work (Table 1). In the following Methods section we introduce both the intermediate products we generate using these data sources, and the final product that is the output of This Study.

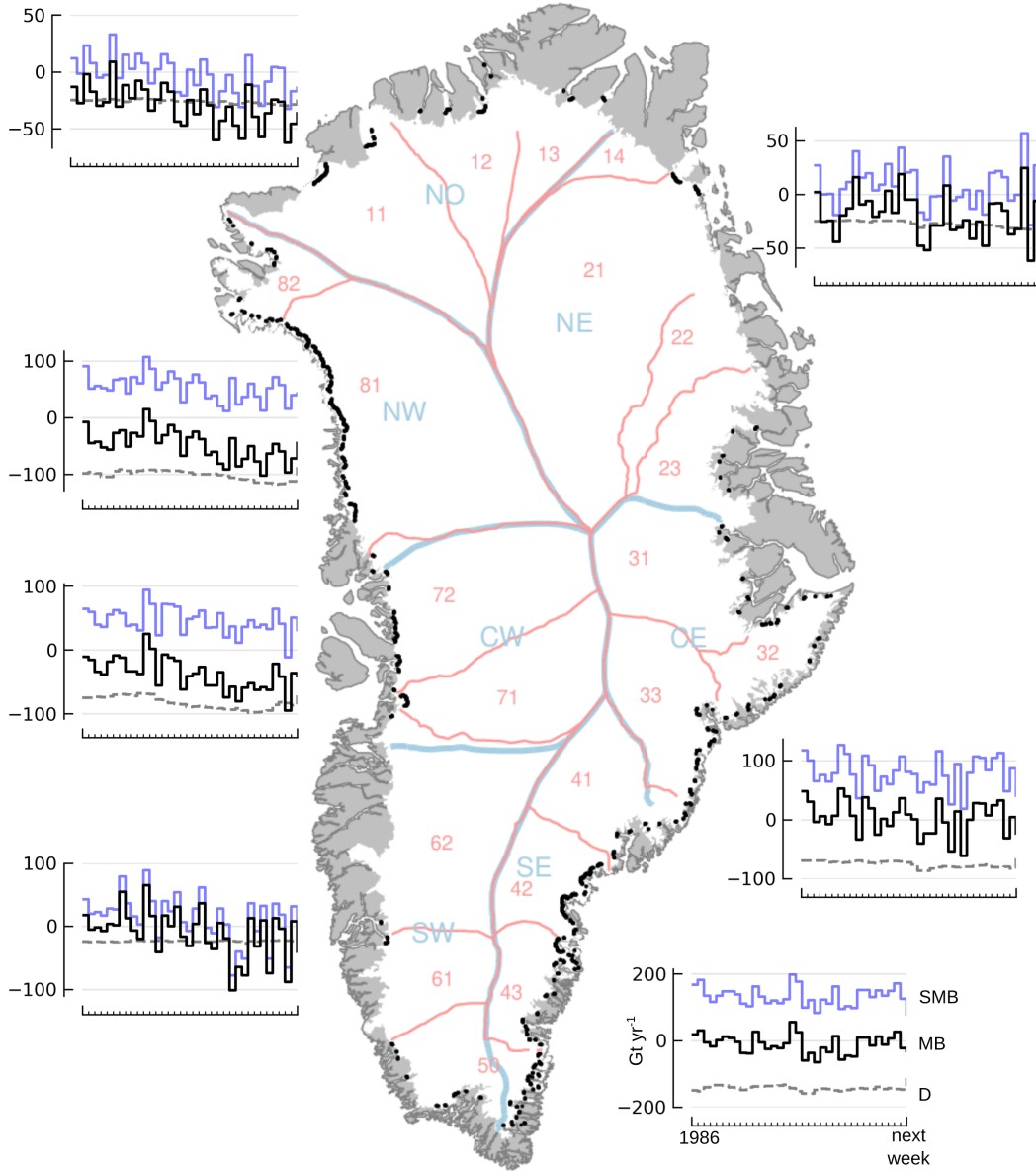

**Figure 1.** Annual mass balance (black lines), surface mass balance (blue lines) and discharge plus basal mass balance (dashed grey) in Gt yr$^{-1}$ for each of the seven Mouginot and Rignot (2019) regions. The map shows both the named regions (Mouginot and Rignot, 2019) and the numbered sectors (Zwally et al., 2012). Discharge gates are marked in black. Only recent (post-1986) data are shown because reconstructed data are not separated into regions or sectors. Next week is defined as `2021-10-11` based on the date this document was compiled.

The inputs to this work are the recent SMB fields from the three RCMs, the recent discharge from Mankoff et al. (2020b) (data: Mankoff and Solgaard (2020)), and the recent basal mass balance fields, of which BMB$_{GF}$ and BMB$_{friction}$ are direct

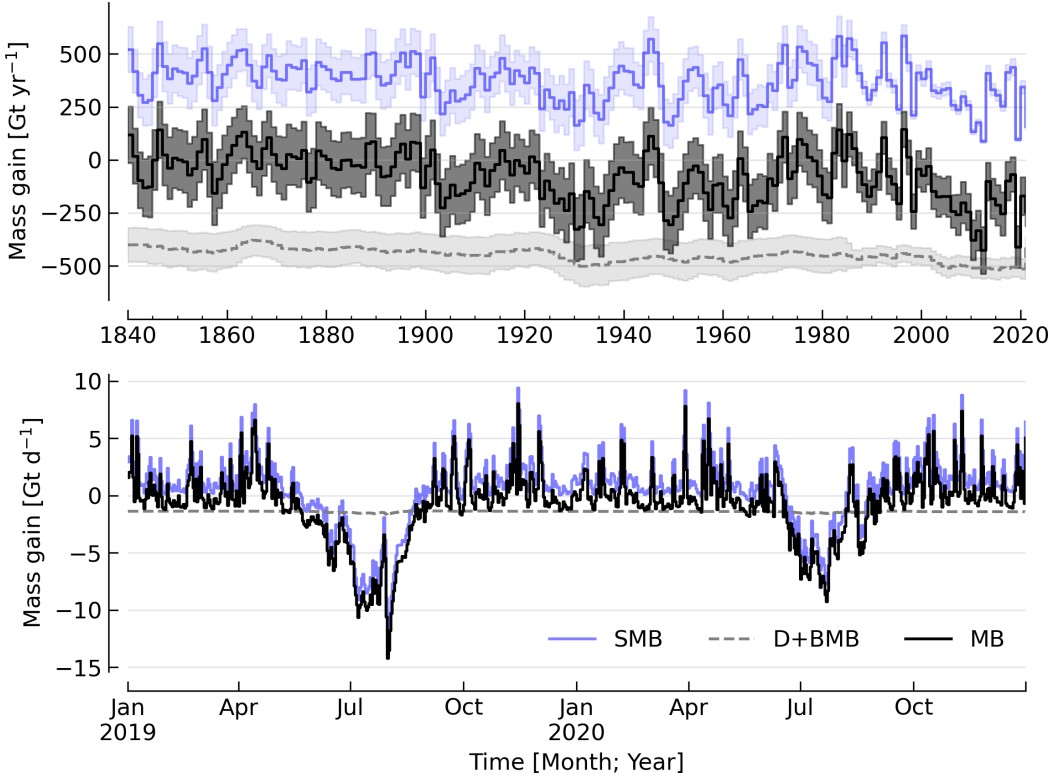

**Figure 2.** Mass balance and its major components. Top panel: Annual average surface mass balance (blue line), discharge (gray dashed), and their mass balance sum (black line). Here the discharge and basal mass balance (D + BMB) are shown with sign inverted (e.g. -1 × (D+BMB)) Lower panel: Same data at daily resolution and limited to 2019 and 2020.

outputs from Karlsson et al. (2021) (data: Karlsson (2021)), but the $BMB_{VHD}$ calculations are redone here (see Methods Sect. 5.3) using the MAR runoff field. The reconstructed data (pre-1986) are surface mass balance and discharge from Kjeldsen et al. (2015) (data: Box et al. (2021)), but adjusted here using the overlapping period (see Methods Sect. 5.4), and runoff from Kjeldsen et al. (2015) (data: Box et al. (2021)) as a proxy and scaled for $BMB_{VHD}$ (see Methods Sect. 5.3).

### 4.1 Surface mass balance

We use one reconstructed SMB from 1840 through 1985, and three recent SMB from 1986 through last month (HIRHAM/HARMONIE, MAR, and RACMO), two through yesterday (HIRHAM/HARMONIE and MAR) and one through next week (MAR).

#### 4.1.1 HIRHAM/HARMONIE

The HIRHAM/HARMONIE product from the Danmarks Meteorologiske Institut (Danish Meteorological Institute; DMI) is based on an offline subsurface firn/SMB model (Langen et al., 2017), which is forced with surface fluxes of energy (turbu-

**Table 1.** Summary of data products used as inputs to This Study.

| Product | Period | Reference | Data / Notes |
|---|---|---|---|
| Reconstructed SMB | 1840 through 1985 | Kjeldsen et al. (2015) | Box et al. (2021) |
| Reconstructed D | 1840 through 1985 | Kjeldsen et al. (2015) | Box et al. (2021) |
| HIRHAM/HARMONIE SMB | 1986 through yesterday | Langen et al. (2017) | |
| MAR SMB | 1986 through next week | Fettweis et al. (2020) | |
| RACMO SMB | 1986 through last month | Noël et al. (2019) | |
| D | 1986 through last month | Mankoff et al. (2020b) | Mankoff and Solgaard (2020) |
| $BMB_{GF}$; $BMB_{friction}$ | 1840 through next week | Karlsson et al. (2021) | Karlsson (2021) |
| $BMB_{VHD}$ | 1840 through 1985 | Kjeldsen et al. (2015) | Box et al. (2021) reconstructed runoff |
| $BMB_{VHD}$ | 1986 through next week | Fettweis et al. (2020) | MAR runoff |

lent and downward radiative) and mass (snow, rain, evaporation, and sublimation). These surface fluxes are derived from the
HIRHAM5 regional climate model for the reconstructed part of the simulation and from DMI's operational numerical weather
forecast model HARMONIE (Iceland-Greenland domain "B", which covers Iceland, Greenland, and the adjacent seas) for the
real-time part. HIRHAM5 is used until 2017-08-31 after which HARMONIE is used.

The HIRHAM5 regional climate model (Christensen et al., 2007) combines the dynamical core of the HIRLAM7 numerical
weather forecasting model (Eerola, 2006) with physics schemes from the ECHAM5 general circulation model (Roeckner et al.,
2003). In the Greenland setup employed here (Lucas-Picher et al., 2012), it has a horizontal resolution of 0.05 ° x 0.05 ° on a
rotated pole grid (corresponding to 5.5 km resolution), and 31 atmospheric levels. It is forced at 6 hour intervals on the lateral
boundaries with horizontal wind vectors, temperature, and specific humidity from the ERA-Interim reanalysis (Dee et al.,
2011). ERA-Interim sea surface temperatures and sea ice concentration are prescribed in ocean grid points. Surface fluxes
from HIRHAM5 are passed to the offline subsurface model.

The offline subsurface model was developed to improve firn details for the HIRHAM5 experiments (Langen et al., 2017).
The subsurface consists of 32 layers with time-varying fractions of snow, ice and liquid water. Layer thicknesses increase with
depth from 6.5 cm water equivalent (w.e.) at the top to 9.2 m w.e. at the bottom giving a full model depth of 60 m w.e. The
processes governing the firn evolution include snow densification, varying hydraulic conductivity, irreducible water saturation
and other effects on snow liquid water percolation, and retention. Runoff is calculated from liquid water in excess of the
irreducible saturation with a characteristic local timescale that depends on surface slope (Zuo and Oerlemans, 1996; Lefebre,
2003). The offline subsurface model is run on the HIRHAM5 5.5 km grid.

HARMONIE (Bengtsson et al., 2017) is a nonhydrostatic model in terrain-following sigma coordinates based on the fully
compressible Euler equations (Simmons and Burridge, 1981; Laprise, 1992). HARMONIE is run at 2.5 km horizontal resolu-
tion and with 65 vertical levels. Compared to previous model versions, upper air 3D variational data assimilation of satellite
wind and radiance data, radio occultation data, radiosonde, aircraft, and surface observations are incorporated. This greatly
improves the number of observations in the model, as in situ observations from ground stations and radiosondes only make

up approximately 20 % of observations in Greenland (Wang and Randriamampianina, 2021; Yang et al., 2018). The model is driven at the boundaries with European Centre for Medium-Range Weather Forecasts (ECMWF) high-resolution data at 9 km resolution. The 2.5 km HARMONIE output is regridded to the 5.5 km HIRHAM grid before input to the offline subsurface model. The HIRHAM5 and the offline model both employ the Citterio and Ahlstrøm (2013) ice mask interpolated to the 5.5 km grid.

### 4.1.2 MAR

The Modèle Atmosphérique Régional (MAR) RCM has been developed by the University of Liège (Belgium) with a focus on the polar regions (Fettweis et al., 2020). The MAR atmosphere module (Gallée and Schayes, 1994) is fully coupled with the soil-ice-snow energy balance vegetation model SISVAT (Gallée et al., 2001) simulating the evolution of the 30 first meters of snow/ice over the ice sheet with the help of 30 snow layers (with time varying thickness) or the 10 first meter of soil over the tundra area. At its lateral boundary, MAR is forced at 6 hour intervals by ERA5 reanalysis and runs at 20 km resolution. The snow pack has been initialised in 1950 from a former MARv3.11 based simulation. Its snow model is based on a former version of the CROCUS snow model (Vionnet et al., 2012) dealing with all the snowpack processes including the meltwater retention, transformation of melting snow and grain size, compaction of snow, formation of ice lenses impacting meltwater penetration, warming of the snowpack from rainfall, and complex snow/bare ice albedo. MAR uses the Greenland Ice Mapping Project (GIMP) ice sheet mask and ice sheet topography (Howat et al., 2014).

We use MAR version 3.12. With respect to version 3.9 intensively validated over Greenland (Fettweis et al., 2020) or the 20 km based MARv3.10 set-up used in Tedesco and Fettweis (2020), MARv3.12 now uses the common polar stereographic projection EPSG 3413. With respect to MARv3.11 fully described in Amory et al. (2021), MARv3.12 assures now the full conservation of water mass into both soil and snowpack at each time step, takes into account of the geographical projection deformations in its advection scheme, better deals with the snow/rain temperature limit with a continuous temperature threshold between 0 and -2°C, increases the evaporation above snow thanks to a saturated humidity computation in SISVAT adapted to freezing temperatures, disallows melt below the 30 m of the resolved snowpack, and includes small improvements and bug fixes with the aim of improving the evaluation of MAR (with both in situ and satellite products) as presented in Fettweis et al. (2020) in addition to small computer time improvements in the parallelisation of its code.

In addition to providing SMB, MAR also provides daily runoff over both permanent ice and tundra area. The ice runoff is used for the daily BMB$_{VHD}$ estimate (Section 5.3).

As the recent SMB decrease (successfully evaluated with GRACE based estimates in Fettweis et al. (2020)) has been fully driven by the increase in runoff (Sasgen et al., 2020), we assume the same degree of accuracy between SMB simulated by MAR (evaluated with the PROMICE SMB database (Fettweis et al., 2020)) and the runoff simulated by MAR.

**Weather-forecasted SMB**: To provide a real-time state of the Greenland ice sheet, MAR is forced automatically every day by the run of 00 h UTC from the Global Forecast System (GFS) model providing weather forecasting initialised by the snow-pack behaviours of the MAR run from the previous day. This continuous GFS forced time series (without any reinitialisation of MAR) provides SMB and runoff estimates between the period covered by ERA5 and the next 7 days. At the end of each

day, ERA5 is used to update the GFS forced MAR time series until about 5 days before the current date and to provide an homogeneous ERA5 forced MAR times series from 1950 to a few days before the current date. We use both the forecasted SMB and forecasted runoff (for $BMB_{VHD}$) fields.

### 4.1.3 RACMO

The Regional Atmospheric Climate MOdel (RACMO) v2.3p2 has been developed at the Koninklijk Nederlands Meteorologisch Instituut (Royal Netherlands Meteorological Institute; KNMI). It incorporates the dynamical core of the High-Resolution Limited Area Model (HIRLAM) and the physics parameterizations of the ECMWF Integrated Forecast System cycle CY33r1. A polar version (p) of RACMO has been developed at the Institute for Marine and Atmospheric research of Utrecht University (UU-IMAU), to assess the surface mass balance of glaciated surfaces. The current version RACMO2.3p2 has been described in detail in Noël et al. (2018), and here we repeat the main characteristics.

The ice sheet has an extensive dry interior snow zone, a relatively narrow runoff zone along the low-lying margins, and a percolation zone of varying width in between. To capture these processes in first order, the original single-layer snow model in RACMO has been replaced by a 40-layer snow scheme that includes expressions for dry snow densification and a simple tipping bucket scheme to simulate meltwater percolation, retention, refreezing, and runoff (Ettema et al., 2010). The snow layers are initialized in September 1957 using temperature and density from a previous run with the offline IMAU Firn Densification Model (Ligtenberg et al., 2018). To simulate drifting snow transport and sublimation, Lenaerts et al. (2012) implemented a drifting snow scheme. Snow albedo depends on snow grain size, cloud optical thickness, solar zenith angle, and impurity content (van Angelen et al., 2012). Bare ice albedo is assumed constant and estimated as the fifth percentile value of albedo time series (2000-2015) from the 500 m resolution MODIS 16-day albedo product (MCD43A3). Minimum/maximum values of 0.30/0.55 are applied to the bare ice albedo, representing ice with high/low impurity content (cryoconite, algae).

To simulate as accurately as possible the contemporary climate and surface mass balance of the ice sheet, the following boundary conditions have been applied. The glacier ice mask and surface topography have been down-sampled from the 90 m resolution Greenland Ice Mapping Project (GIMP) digital elevation model (Howat et al., 2014). At the lateral boundaries, model temperature, specific humidity, pressure, and horizontal wind components at the 40 vertical model levels are relaxed towards 6-hourly ECMWF reanalysis (ERA) data. For this we use ERA-40 between 1958 and 1978 (Uppala et al., 2005), ERA-Interim between 1979 and 1989 (Dee et al., 2011), and ERA-5 between 1990 and 2020 (Hersbach et al., 2020). The relaxation zone is 24 grid cells (~130 km) wide to ensure a smooth transition to the domain interior. This run has active upper atmosphere relaxation (van de Berg and Medley, 2016). Over glaciated grid points, surface aerodynamic roughness is assumed constant for snow (1 mm) and ice (5 mm). In this run, RACMO2.3p2 has 5.5 km horizontal resolution over Greenland and the adjacent oceans and land masses, but it was found previously that this is insufficient to resolve the many narrow outlet glaciers. The 5.5 km product is therefore statistically downscaled onto a 1 km grid sampled from the GIMP DEM (Noël et al., 2019), employing corrections for biases in elevation and bare ice albedo using a MODIS albedo product at 1 km resolution (Noël et al., 2016).

### 4.1.4 Reconstructed

The Kjeldsen et al. (2015) 173-year (1840 through 2012) mass balance reconstruction is based on the Box (2013) 171-year (1840 through 2010) statistical reconstruction. Kjeldsen et al. (2015) add a more sophisticated meltwater retention scheme (Pfeffer et al., 1991); weighting of in situ records in their contribution to the estimated value; dispersal of annual accumulation to monthly; and extend the reconstruction in time through 2012.

The Box (2013) 171-year (1840-2010) reconstruction is developed from linear regression parameters that describe the least 190 squares regression between a) spatially discontinuous in situ monthly air temperature records (Cappelen et al., 2011; Cappelen, 2001; Cappelen et al., 2006; Vinther et al., 2006)) or firn/ice cores (Box et al., 2013) and b) spatially continuous outputs from regional climate model RACMO version 2.1 (Ettema et al., 2010). A 43-year overlap period (1960 through 2012) with the RACMO data are used to determine regression parameters (slope, intercept) on a 5 km grid cell basis. Temperature data define melting degree days, which have a different coefficient for bare ice than snow cover, determined from hydrological-195 year cumulative SMB. A fundamental assumption is that the calibration factors, regression slope, and offset for the calibration period 1960 through 2012 are stationary over time for which there is some evidence of in Fettweis et al. (2017). Box et al. (2013) describes the methods in more detail.

The reconstructed surface mass balance is adjusted as described in the Methods Sect. 5.4 (Fig. 3).

### 4.2 Discharge

The recent discharge data are from Mankoff et al. (2020b) (data: Mankoff and Solgaard (2020)). This product covers all fast-flowing (> 100 m yr$^{-1}$) marine-terminating glaciers. The discharge in Mankoff et al. (2020b) is computed at flux gates ~5 km upstream from glacier termini (Mankoff, 2020), using a wide range of velocity products, and ice thickness from BedMachine v4. Discharge across flux gates is derived with a 200 m spatial resolution grid, but then summed and provided at glacier resolution. Temporal coverage begins in 1986 with a few velocity estimates, and is updated each time a new velocity product 205 is released, which is every ~12 days with a ~30 day lag (Solgaard et al. (2021); data: Solgaard and Kusk (2021)).

Some changes have been implemented since the last publication describing the discharge product (i.e., Mankoff et al. (2020b)). These are minor and include updating the Khan et al. (2016) (data: Khan (2017)) surface elevation change product from 2015 through 2019, updating various MEaSUREs velocity products to their latest version, updating the PROMICE Sentinel ice velocity product from Edition 1 (doi:10.22008/promice/data/sentinel1icevelocity/greenlandicesheet/v1.0.0) to Edi-210 tion 2 (Solgaard et al. (2021); Solgaard and Kusk (2021)), and updating from BedMachine v3 (supplemented in the SE with Millan et al. (2018)) to use only BedMachine v4 (Morlighem et al., 2021).

The reconstructed discharge data (Kjeldsen et al., 2015) are estimated via a linear fit between unsmoothed annual discharge spanning 2000 to 2012 (Enderlin et al., 2014) and runoff data from (Kjeldsen et al., 2015) using a 6-year trailing average. The method for scaling discharge from runoff was introduced by (Rignot et al., 2008), who scaled the SMB anomaly with discharge. 215 Sensitivity analyses conducted by Box and Colgan (2013) showed runoff to be the more effective discharge predictor, and include a discussion of the physical basis. Although the fitting period of the present dataset includes an anomalous period

of discharge (2000 through 2005; e.g., Boers and Rypdal (2021)), the discharge data used by Rignot et al. (2008) and Box and Colgan (2013) also includes years 1958 and 1964 that lie near the regression line (See Box and Colgan (2013) Fig. 4 and related section 4. Physical basis). Further, while 2000 through 2005 cover a changing period in Greenlandic discharge (Mankoff et al., 2020b; King et al., 2020), there were likely other anomalous periods in the past, when glaciers in Greenland experienced considerable increases in discharge as inferred by geological and geodetic investigations (Andresen et al., 2012; Bjørk et al., 2012; Khan et al., 2015, 2020).

The reconstructed discharge is adjusted as described in the Methods Sect. 5.4.

## 4.3 Basal mass balance

The basal mass balance (BMB; Karlsson et al. (2021)) comes from mass lost at the bed from geothermal flux (BMB$_{GF}$), frictional heating (BMB$_{friction}$) from the basal shear velocity, and viscous heat dissipation (BMB$_{VHD}$) from surface runoff routed to the bed (i.e. the volume of the subglacial conduits formed from surface runoff; Mankoff and Tulaczyk (2017)).

These fields (data: Karlsson (2021)) are provided as steady state annual estimates. We use the BMB$_{GF}$ and BMB$_{friction}$ products and apply 1/365th to each day, each year. Because BMB$_{VHD}$ is proportional to runoff, an annual estimate is not appropriate for this work with daily resolution. We therefore re-calculate the BMB$_{VHD}$-induced basal melt as described in Methods Sect. 5.3.

### 4.3.1 Geothermal Flux

Due to a lack of direct observations, the geothermal flux is poorly constrained under most of the Greenland ice sheet. Different approaches have been employed to infer the value of the BMB$_{GF}$ often with diverging results (see e.g., Rogozhina et al. (2012); Rezvanbehbahani et al. (2019)). Lacking substantial validation that favours one BMB$_{GF}$ map over the others, Karlsson et al. (2021) instead use the average of three widely used BMB$_{GF}$ estimates: Fox Maule et al. (2009); Shapiro and Ritzwoller (2004), and Martos et al. (2018). The BMB$_{GF}$ melt rate is calculated as

$$\dot{b}_m = E_{GF}\,\rho_i^{-1}\,L^{-1}, \tag{1}$$

where $E_{GF}$ is available energy at the bed, here the geothermal flux in units W m$^{-2}$, $\rho_i$ is the density of ice (917 kg m$^{-3}$), and $L$ is the latent heat of fusion (335 kJ kg$^{-1}$; Cuffey and Paterson (2010)). BMB$_{GF}$ melting is only calculated where the bed is not frozen. We use the MacGregor et al. (2016) estimate of temperate bed extent and scale Eq. 1 by 0, 0.5, or 1 where the bed is frozen (~25 % of the ice sheet area), uncertain (~33 %), or thawed (~42 %), respectively.

### 4.3.2 Friction

This heat term stems from the friction produced as ice slides over the bedrock. The term has only been measured in a handful of places (e.g., Ryser et al. (2014); Maier et al. (2019)) and it is unclear how representative those measurements are at ice-sheet scales. Karlsson et al. (2021) therefore estimate the frictional heating using the Full Stokes Elmer/Ice model that resolves all

stresses while relating basal sliding and shear stress using a linear friction law (Gillet-Chaulet et al., 2012; Maier et al., 2021). The model is tuned to match a multi-decadal surface velocity map (Joughin et al., 2018) covering 1995-2015 and it returns an estimated basal friction heat that is used to calculate the basal melt due to friction, similar to Eq. 1:

$$\dot{b}_m = E_f \, \rho_i^{-1} \, L^{-1}, \tag{2}$$

where $E_f$ is energy due to friction. We also apply the 0, 0.5, and 1 scale as used for the $BMB_{GF}$ term (MacGregor et al., 2016) in order to mask out areas that are likely frozen.

### 4.4 Other

ROI regions come from Mouginot and Rignot (2019) and ROI sectors come from Zwally et al. (2012).

### 4.5 Products used for validation

We validate This Study against five other data products (See Table 2 and Sect. 6). These products are the most recent IO product (Mouginot et al., 2019), the previous PROMICE mass balance product (Colgan et al. (2019); data: Colgan (2021)), the two mostly-independent methods of estimating ice sheet mass change: GMB (Barletta et al. (2013); data: Barletta et al. (2020)) and VC (Simonsen et al. (2021a); data: Simonsen et al. (2021b)), and the IMBIE2 data (The IMBIE Team, 2019). In addition to this we evaluate the reconstructed Kjeldsen et al. (2015) (data: Box et al. (2021)) and This Study data during the overlapping period 1986 through 2012.

## 5 Methods

The total mass balance for all of Greenland and all the different ROIs involves summing each field (SMB, D, BMB) by each ROI, then subtracting the D and BMB from the SMB fields, or,

$$MB = SMB - D - BMB. \tag{3}$$

Products that do not include the BMB term (i.e., Mouginot et al. (2019); Colgan et al. (2019), and Kjeldsen et al. (2015)) have total mass balance defined as

$$MB^* = SMB - D, \tag{4}$$

and when comparing This Study to those products, we compare like terms, never comparing our MB to a different product $MB^*$, except in Fig. 4 where all products are shown together.

Prior to calculating the mass balance, we perform the following steps.

## 5.1 Surface mass balance

In This Study we generate an output based on each of the three RCMs (HIRHAM/HARMONIE, MAR, and RACMO), however, in addition to these we generate a final and 4th SMB field defined as a combination of 1) the adjusted reconstructed SMB from 1840 through 1985 (Sect. 5.4), and 2) the average of HIRHAM/HARMONIE, MAR, and RACMO from 1986 through a few months ago, the average of HIRHAM/HARMONIE and MAR from a few months ago through yesterday, and MAR from yesterday through next week. See the Appendix A for differences among This Study MB and MB derived using each of the RCM SMBs. There is no obvious change or step function at the 1985 to 1986 reconstructed-to-recent change, nor as the RACMO and then HIRHAM/HARMONIE RCMs become unavailable a few months ago and yesterday, respectively.

## 5.2 Projected discharge

We project the discharge from the last observed point from Mankoff et al. (2020b) (generally between 2 weeks and 1 month old) to seven days into the future at each glacier. We define the long-term trend as the linear least squares fit to the last three years of data. The residual is the data minus the long-term trend. We define the seasonal signal as the daily average from each year of the last three years of the residual during the temporal window of interest that spans from the most recently available observation through next week. We shift the seasonal signal so that it is 0 on the first projected day. We then assign the value of the last observation, plus the long term trend, plus the seasonal signal to the recent past projected and future forecasted D.

Discharge does not change sign and changes magnitude by approximately 6 % annually over the entire ice sheet (King et al., 2018), but surface mass balance changes sign and has both larger and higher frequency variability. From this, the statistical forecast for discharge described above does not impact results as much as the physically-based model forecast for surface mass balance.

## 5.3 Basal mass balance

Because Karlsson et al. (2021) provide a steady-state annual-average estimate of the BMB fields, we divide the $BMB_{GF}$ and $BMB_{friction}$ fields by 365 to estimate daily average. This is a reasonable treatment of the $BMB_{GF}$ field, which does not have an annual cycle. The $BMB_{friction}$ field does have a small annual cycle that matches the annual velocity cycle. However, when averaged over all of Greenland, this is only a ~6 % variation (King et al., 2018), and Karlsson et al. (2021) found that basal melt rates are 5 % higher during the summer. Thus, the intra-annual changes are less than the uncertainty. The $BMB_{VHD}$ field varies significantly throughout the year, because it is proportional to surface runoff. We therefore generate our own $BMB_{VHD}$ for this study.

To estimate recent $BMB_{VHD}$ we use daily MAR runoff (see Mankoff et al. (2020a)) and BedMachine v4 (Morlighem et al., 2017, 2021) to derive subglacial routing pathways, similar to Mankoff and Tulaczyk (2017). We assume that all runoff travels to the bed within the grid cell where it is generated, the bed is pressurized by the load of the overhead ice, and the runoff

discharges on the day it is generated. We calculate subglacial routing from the gradient of the subglacial pressure head surface, $h$, defined as

$$h = z_b + k \frac{\rho_i}{\rho_w}(z_s - z_b), \tag{5}$$

with $z_b$ the basal topography, $k$ the flotation fraction (1), $\rho_i$ the density of ice (917 kg m$^{-3}$), $\rho_w$ the density of water (1000 kg m$^{-3}$), and $z_s$ the ice surface. Eq. 5 comes from Shreve (1972), where the hydropotential has units of pascals (Pa), but here it is divided by gravitational acceleration $g$ times the density of water $\rho_w$ to convert the units from pascals to meters (Pa to m).

We compute $h$ and from $h$ streams and outlets, and both the pressure and elevation difference between the source and outlet. The energy available for basal melting is the elevation difference (gravitational potential energy) and two-thirds of the pressure

difference, with the remaining one third consumed to warm the water to match the changing phase transition temperature (Liestøl, 1956; Mankoff and Tulaczyk, 2017). We assume all energy, $E_{\mathrm{VHD}}$ (in Joules), is used to melt ice with

$$b_m = E_{\mathrm{VHD}}\, \rho_i^{-1}\, L^{-1}. \tag{6}$$

Because results are presented per ROI and to reduce the computational load of this daily estimate, we only calculate the integrated energy released between the RCM runoff source cell and the outlet cell, and then assign that to the ROI containing

315 the runoff source cell.

To estimate reconstructed basal mass balance, we treat BMB$_{\mathrm{GF}}$ and BMB$_{\mathrm{friction}}$ as steady state as described at the start of this section. For BMB$_{\mathrm{VHD}}$ we use the fact that VHD comes from runoff by definition, and from this, reconstructed BMB$_{\mathrm{VHD}}$ is calculated using scaled runoff as a proxy. VHD theory suggests that a unit volume of runoff that experiences a 1000 m elevation drop will release enough heat to melt an additional 3 % (Liestøl, 1956). To estimate the scale factor we use the 1986 through

2012 overlap between Kjeldsen et al. (2015) runoff and This Study recent BMB$_{\mathrm{VHD}}$ from MAR runoff described above. The correlation between the two has an r$^2$ value of 0.75, slope of 0.03, and an intercept of -3 Gt yr$^{-1}$ (Appendix D). From this, we scale the Kjeldsen et al. (2015) reconstructed runoff by 3 % (from the 0.03 slope, unrelated to the theoretical 1000 m drop described earlier) to estimate reconstructed BMB$_{\mathrm{VHD}}$.

## 5.4 Reconstructed adjustment

We use the reconstructed and recent surface mass balance (SMB) and discharge (D) overlap from 1986 through 2012 to adjust the reconstructed data. This Study vs reconstructed SMB has a slope of 0.6 and an intercept of 166 Gt yr$^{-1}$ (Fig. 3 SMB), and This Study vs reconstructed D has a slope of 1.1 and an intercept of -17 Gt yr$^{-1}$ (Fig. 3 D). The unadjusted reconstructed data slightly underestimates years with high SMB and overestimates years with low SMB (see 1986, 2010, 2011, and 2012 in Fig. 3 SMB). The unadjusted reconstructed data slightly overestimates years with low D and overestimates years with high D.

We adjust the reconstructed data until the reconstructed vs. recent slope is 1 and intercept is 0 Gt yr$^{-1}$ for each of the surface mass balance and discharge comparisons (Fig. 3). We then derive the BMB$_{\mathrm{VHD}}$ term for reconstructed basal mass balance

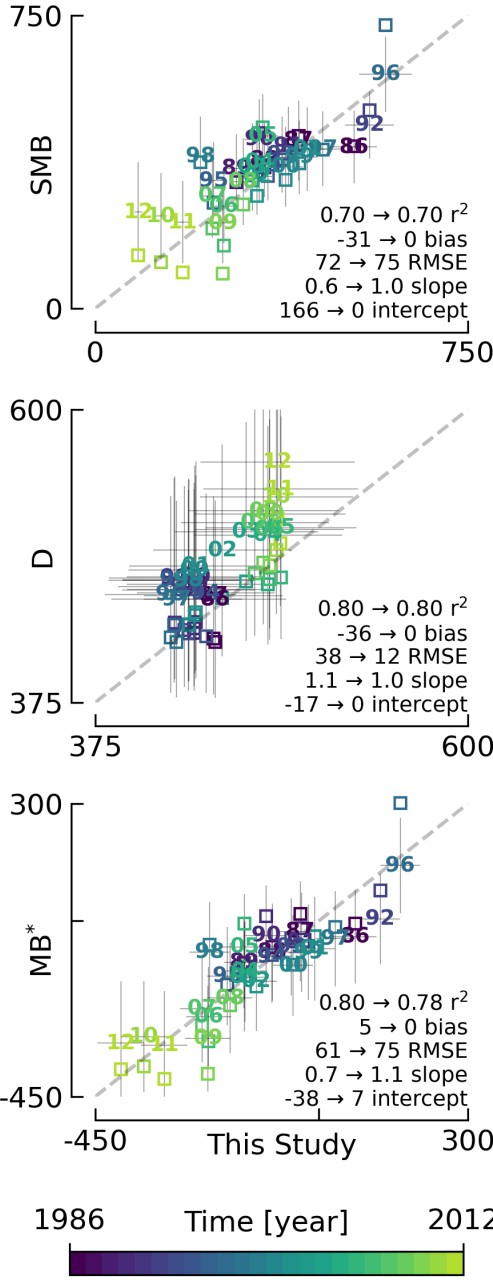

**Figure 3.** Comparison between This Study and the reconstructed (Kjeldsen et al., 2015). All axes units are Gt yr$^{-1}$. Plotted numbers represent the last two digits of the years for the unadjusted data sets. The matching colored squares show the adjusted data. MB$^*$ shown here does not include BMB for either the reconstructed or This Study data. Arrows show statistical properties before and after the adjustment. No adjustment is made to MB$^*$, but it is computed from Eq. 4 both before (numbered) and after (squares) the surface and discharge adjustments.

(Sect. 5.3 and Appendix D), bring in the other BMB terms (Sect. 5.3), and use Eq. 3 to compute the adjusted reconstructed mass balance.

For reconstructed SMB and D, the mean of the recent uncertainty is added to the reconstructed uncertainty during the adjustment. Reconstructed MB uncertainty is then re-calculated as the square root of the sum of the squares of the reconstructed SMB and D uncertainty.

For surface mass balance, the adjustment is effectively a rotation around the mean values, with years with low SMB decreasing and years with high SMB increasing after the adjustment. For discharge, years with low D are slightly reduced, and years with high D have a higher reduction to better match the overlapping estimates.

The adjustment described above treats all biases in the reconstructed data. The primary assumption of our adjustment is that the bias contributions do not change in proportion to each other over time. We attribute the disagreement and need for the adjustment to the demonstrated too-high biases in accumulation and ablation estimates in the 1840-2012 reconstructed SMB field (Fettweis et al., 2020), an offset resulting from differences in ice masks (Kjeldsen et al., 2015), the inclusion of peripheral glaciers (Kjeldsen et al., 2015), other accumulation rate inaccuracies (Lewis et al., 2017, 2019), and other unknowns.

## 5.5 Domains, boundaries, and regions of interest

Few of the ice masks used here are spatially aligned. The Zwally et al. (2012) sectors and the Mouginot and Rignot (2019) regions are often smaller than the RCM ice domains. For example, the RACMO ice domain is 1,718,959 km$^2$, of which 1,696,419 km$^2$ (99 %) are covered by the Mouginot and Rignot (2019) regions, and 22,540 km$^2$ (1 %) are not, or 1,678,864 km$^2$ (98 %) are covered by the Zwally et al. (2012) and 40,095 km$^2$ (2 %) are not.

Cropping the RCM domain edges would remove the edge cells where the largest SMB losses occur. This effect is minor when SMB is high (years with low runoff, assuming SMB magnitude is dominated by the runoff term). This effect is large when SMB is low (years with high runoff). As an example for the 2010 decade, RACMO SMB has a mean of 251 Gt yr$^{-1}$ for the decade, with a low of 45 Gt in 2019, a high of 420 Gt in 2018. For these same extreme years RACMO cropped to Mouginot and Rignot (2019) has a low of 76 Gt (68 % high) and a high of 429 Gt (2 % high). RACMO cropped to Zwally et al. (2012) has a low of 84 Gt (85 % high) and a high of 429 Gt (2 % high).

We therefore grow the ROIs to cover the RCM domains. ROIs are grown by expanding them outward, assigning the new cells the value (ROI classification, that is sector number or region name, see Fig. 1) of the nearest non-null cell, and then clipping to the RCM ice domain. This is done for each ROI and RCM. Appendix E provides a graphical display of the HIRHAM RCM domain, the Mouginot and Rignot (2019) domain, and our expanded Mouginot and Rignot (2019) domain.

BMB$_{VHD}$ comes from the MAR ice domain runoff, but is generated on the BedMachine ice thickness grid, which is smaller than the ice domain in some places. Therefore, the largest runoff volumes per unit area (from the low-elevation edge of the ice sheet) are discarded in these locations.

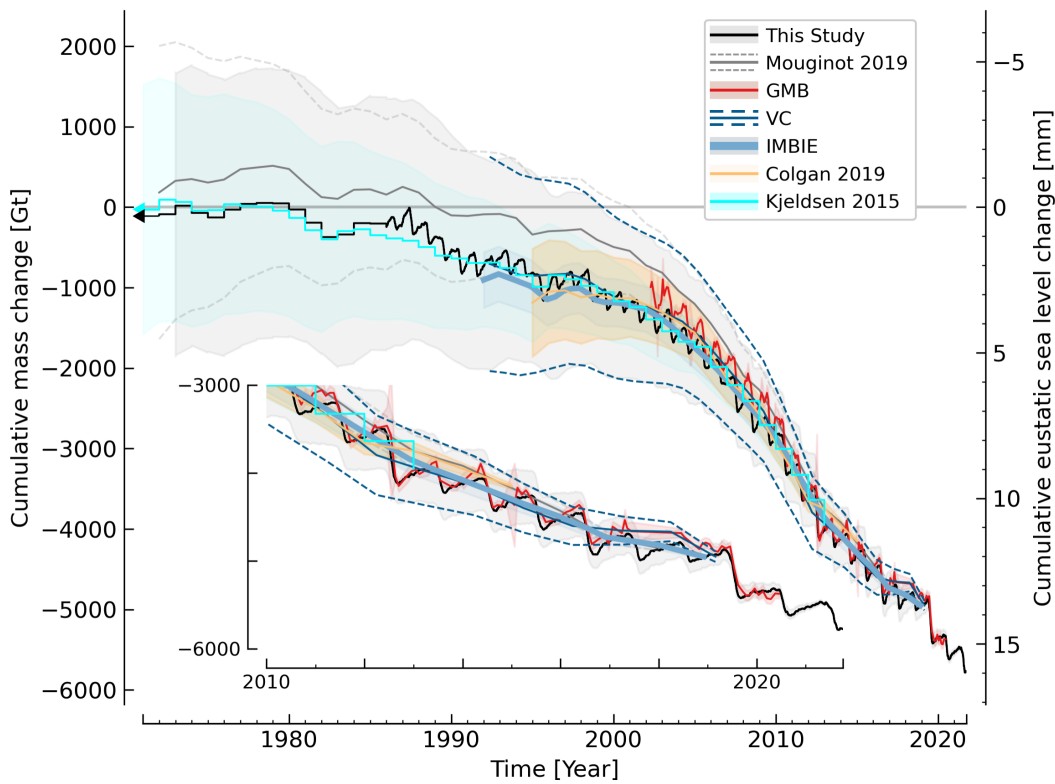

**Figure 4.** Comparison between This Study and other mass balance time series. Note that various products do or do not include basal mass balance or peripheral ice masses (see Table 2). This Study annual-resolution data prior to 1986 is the Kjeldsen et al. (2015) data adjusted as described in Sect. 5.4. Sea level rise calculated as -Gt/361.8. Inset highlights changes since 2010. Data product version 50 from `2021-10-04` used to generate this graphic.

## 6 Product evaluation and assessment

We compare to six related data sets (see Table 2 and Sect. 4.5): The most similar and recent IO product (Mouginot et al., 2019), the previous PROMICE assessment (Colgan et al., 2019), the two mostly independent methods (GMB (Barletta et al., 2013) and VC (Simonsen et al., 2021a)), IMBIE2 (The IMBIE Team, 2019), and the unadjusted reconstructed/recent overlap (Kjeldsen et al., 2015).

Our initial comparison (Fig. 4) shows all seven products overlaid in a time series accumulating at the product resolution (daily to annual) from the beginning of the first overlap (1972, Mouginot et al. (2019)) until seven days from now (now defined as `2021-10-04` based on the date this document is compiled). Each data set is manually aligned vertically so that the last timestamps appear to overlap, allowing disagreements to grow back in time. We also assume errors are smallest at present and allow errors to grow back in time. The errors for this product are described in the Uncertainty section.

In the sections below, we compare This Study to each of the validation data in more detail. The Mouginot et al. (2019) and Colgan et al. (2019) products allow term-level (SMB, D, and MB$^*$) comparison, and the GMB, VC, and IMBIE2 only MB-level comparison. The MB or MB$^*$ comparison for each product is summarized in Table 2. All have different masks. Bias [Gt yr$^{-1}$] is defined as $\frac{1}{n}\sum_{i=1}^{n}(x_i - y_i)$. RMSE [Gt yr$^{-1}$] is defined as $\sqrt{\frac{1}{n}\sum_{i=1}^{n}(x_i - y_i)^2}$. Sums are computed using ice-sheet wide annual values, where $x$ is This Study, $y$ is the other product, and a positive bias means that This Study has a larger value.

**Table 2.** Summary of correlation, bias, and RMSE between different products during their overlap periods with This Study. Basal mass balance not included in This Study when comparing against Mouginot and Rignot (2019), Colgan et al. (2019), or Kjeldsen et al. (2015). Peripheral ice masses never included in This Study.

| Other product | $r^2$ | bias | RMSE | Fig. | Overlap | Notes |
|---|---|---|---|---|---|---|
| Mouginot et al. (2019) | 0.94 | 11 | 38 | 5 | 1986 – 2018 | No basal mass balance |
| Colgan et al. (2019) | 0.87 | -32 | 59 | 6 | 1995 – 2015 | No basal mass balance |
| GMB | 0.86 | 32 | 63 | 7 | 2002 – 2020 | Includes peripheral masses |
| VC | 0.62 | -11 | 86 | 7 | 1992 – 2019 | Multi-year smooth |
| IMBIE2 | 0.89 | -7 | 44 | 7 | 1992 – 2018 | No BMB when using IO; BMB when using GMB or VC |
| Kjeldsen et al. (2015) | 0.80 | 5 | 61 | 3 | 1986 – 2012 | No basal mass balance; Includes peripheral masses |

## 6.1 Mouginot (2019)

The Mouginot et al. (2019) product spans the 1972 through 2018 period. We only use 1986 and onward because This Study has annual resolution prior to 1986 and Mouginot et al. (2019) data are provided on a non-calendar year period. The SMB comes from RACMO v2.3p2 downscaled at 1 km, and agrees very well with SMB from This Study ($r^2$ `0.94`, bias `11`, RMSE `38`, slope 1.1). The minor SMB differences are likely due to mask differences, or our use of a three-RCM average SMB estimate.

Mouginot et al. (2019) discharge and our D from Mankoff et al. (2020b) have a -33 Gt yr$^{-1}$ bias. This difference can mainly be attributed to different discharge estimates in the Southeast and Central east sector (Appendix: Mouginot regions). When we include BMB in This Study (diamonds in middle panel shifting values to the right), it adds ~25 Gt yr$^{-1}$ to This Study.

Because MB$^*$ is a linear combination of SMB and D terms (Eq 4), the MB$^*$ differences between this product and Mouginot et al. (2019) are dominated by the D term, although it is not apparent because interannual variability is dominated by SMB.

## 6.2 Colgan (2019)

The Colgan et al. (2019) product spans 1995 through 2015. The SMB term is broadly similar to the RCM-averaged SMB term in This Study, although Colgan et al. (2019) use only an older version of MAR (Fig. 6 top panel). The Colgan et al. (2019) SMB is spatially interpolated over the PROMICE ice-sheet ice mask (Citterio and Ahlstrøm, 2013), which contains more detail on the ice sheet periphery, and therefore a larger ablation area than the native coarser MAR ice mask. This Study does not interpolate the SMB field and instead works on the SMB ice domain.

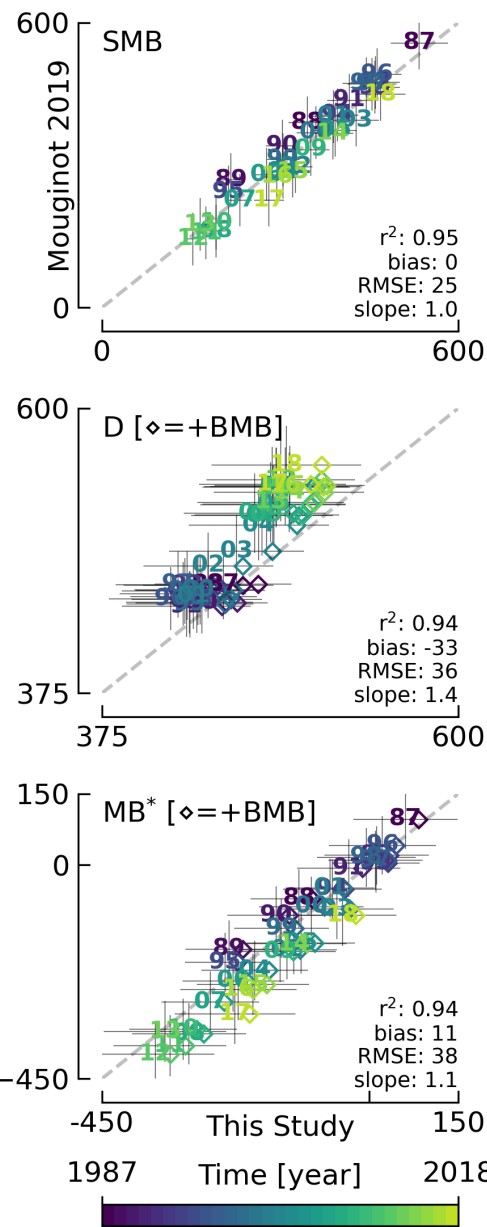

**Figure 5.** Comparison of This Study vs Mouginot et al. (2019). All axes units are Gt yr$^{-1}$. Plotted numbers represent the last two digits of the year. Matching colored diamonds show the data when BMB is added to This Study. Printed numbers (r$^2$, bias, RMSE, slope) compare values without BMB.

The largest difference between This Study and Colgan et al. (2019) is that the latter estimates grounding line ice discharge
based on corrections to ice volume flow rate measured across the ~1700 m elevation contour. This is far inland relative to the

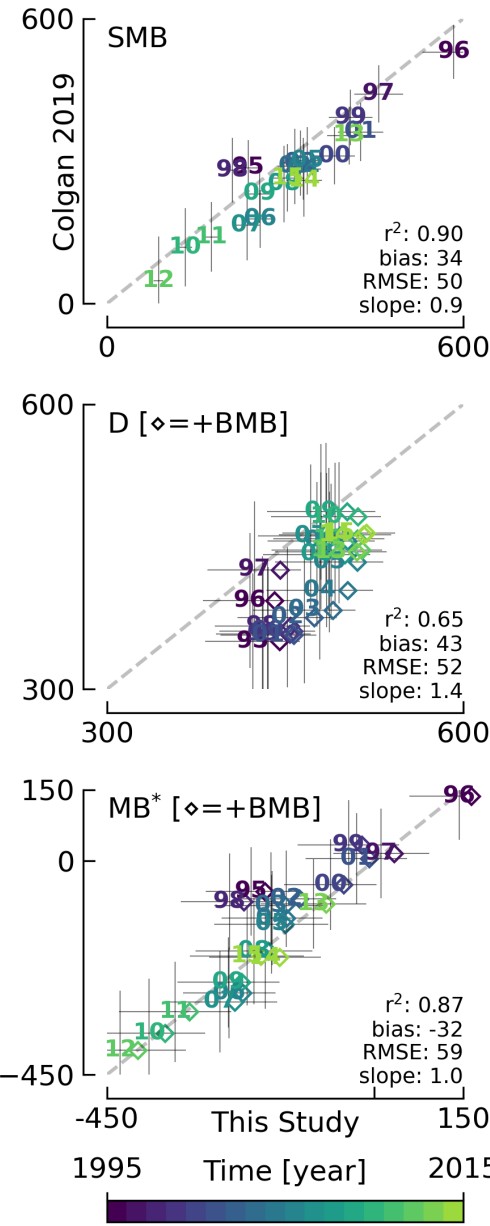

**Figure 6.** Comparison of This Study vs Colgan et al. (2019). All axes units are Gt yr$^{-1}$. Plotted numbers represent the last two digits of the year. Matching colored diamonds show the data when BMB is added to This Study. Printed numbers (r$^2$, bias, RMSE, slope) compare values without BMB.

grounding line flux gates used in This Study (from Mankoff (2020)). This introduces uncertainty in the Colgan et al. (2019) D term from SMB corrections between the 1700 m elevation contour and the terminus (see large disagreement in Fig. 6 mid

panel). This disagreement increases when BMB is included in the results of This Study (shown by the annual values shifting to the right).

The D disagreement is represented differently across sectors (Appendix: Colgan 2019), where sectors 1, 2, 5, and 6 all have correlation coefficients less than ~0.1, while the remaining sectors 3, 4, 7, and 8 all have correlation coefficients greater than 0.5.

     This Study assesses greater D bias (43 Gt yr$^{-1}$) than Colgan et al. (2019). While Colgan et al. (2019) did not assess BMB, the majority of this discrepancy likely results from Colgan et al. (2019) aliasing the aforementioned downstream correction terms.

For example, while This Study shows very little interannual variability in ice discharge in the predominantly land-terminating SW region, Colgan et al. (2019) infer large interannual variability in ice discharge based on large interannual variability in SMB and changes in ablation area ice volume in their Sector 6. The discrepancy between This Study and Colgan et al. (2019) D [+BMB] is largest during the earliest part of the record (i.e. 1995-2000), decreasing towards present-day, which may suggest that Colgan et al. (2019) particularly overestimated the response in ice discharge to 1990s climate variability.

Similar to the comparison with Mouginot et al. (2019), the disagreement between This Study and Colgan et al. (2019) is dominated by D disagreement, although it is again not apparent because interannual variability is dominated by SMB.

## 6.3    Gravimetric Mass Balance (GMB)

Unlike this Study, the GMB method includes mass losses and gains on peripheral ice masses which should introduce a bias of ~10 to 15% (Colgan et al., 2015; Bolch et al., 2013). The inclusion of peripheral ice in the GMB product is because the

415 spatial resolution is so low that it cannot distinguish between them and the main ice sheet. There is also signal leakage from other glaciated areas, e.g., the Canadian Arctic. This can have an effect on the estimated signal, especially in sectors 1 and 8 or regions NW and NO. There is also leakage between basins, which becomes a larger issue for smaller basins or where major outlet glaciers are near basin boundaries. GMB may also have an amplified seasonal signal due to changing snow loading in the surrounding land areas that may be mapped as ice sheet mass change variability. This would enhance the seasonal amplitude but

not have an impact on the interannual mass change rates. Additionally, different glacial isostatic adjustment (GIA) corrections applied to the gravimetric signal may also lead to differences in GMB estimates on ice sheet scale, but also on sector scale (e.g. Sutterley et al. (2014); Khan et al. (2016)).

     GMB and the IO method (This Study) both report changes in ice sheet mass, but they are measuring two fundamentally different things. The IO method tracks volume flow rate across the ice sheet boundaries. Typically this is meltwater across

the ice sheet surface and solid ice across flux gates near the calving edge of the ice sheet, and in This Study also meltwater across the ice sheet basal boundary. That volume is then converted to mass. We consider that mass is 'lost' as soon as it crosses the boundary (i.e. the ice melts or ice crosses the flux gate). The GMB method tracks the regional mass changes. Melting ice has no impact on this, until the meltwater enters the ocean and a similar mass leaves the far-field GMB footprint. From these differences, the GMB method may be a better estimate of sea level rise, while the IO method may be a better representation of

the state of the Greenland ice sheet.

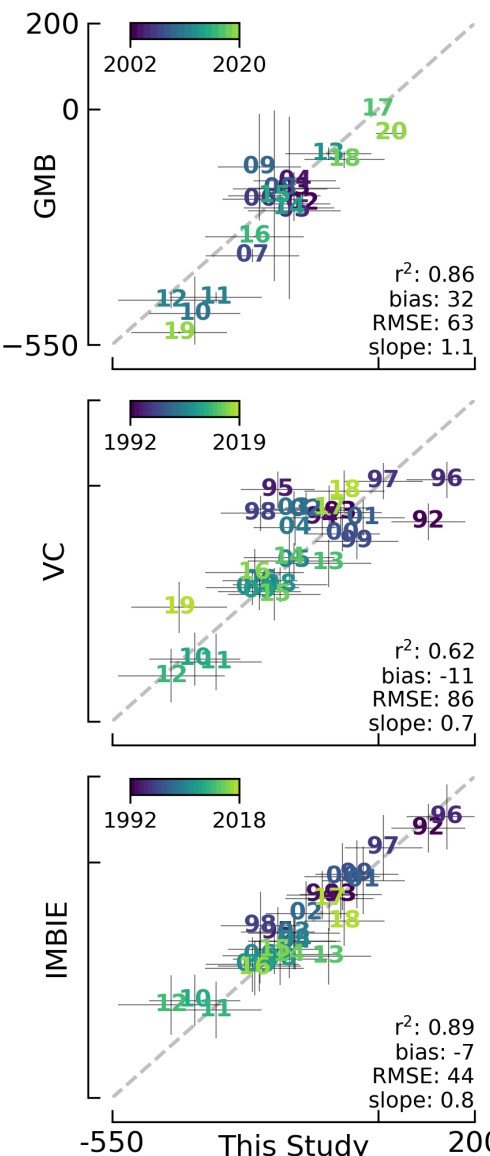

**Figure 7.** This Study total mass balance (MB) vs. the gravimetric method (GMB), volume change method (VC) and IMBIE2 estimates of MB. All three include BMB. All axes units are Gt yr$^{-1}$. Plotted numbers represent the last two digits of the year. GRACE and IMBIE2 include peripheral ice masses.

## 6.4 Volume Change (VC)

When deriving surface elevation change from satellite altimetry, data from multiple years are needed to give a stable ice sheet-wide prediction. Hence, the altimetric mass balance estimates are often reported as averages of single satellite missions.

Although This Study has a small ($-11$ Gt yr$^{-1}$) bias in comparison to Simonsen et al. (2021a) VC, there is a relatively high
RMSE of $86$ Gt yr$^{-1}$ and a mid-range correlation ($r^2 = 0.62$). This suggests that while both This Study and VC agree on the
total mass loss of the ice sheet, they disagree on the precise temporal distribution of this mass loss. It is possible the outlying
1992 and 2019 years are influenced by the edge of the time series record if not fully sampled, but other outliers exist - the 1992
extreme low melt year and the 2019 extreme melt year, as well as the 1995 through 1998 period, stand out as years with poor
agreement.

We suggest that this is due to climate influences on the effective radar horizon across the ice sheet during these years.
Weather-driven changes in the effective scatter horizon, mapped by Ku-band in the upper snow layer of ice sheets hampers the
conversion of radar-derived elevation change into mass change (Nilsson et al., 2015). Simonsen et al. (2021a) used a machine
learning approach to derive a temporal calibration field for converting the radar elevation change estimates into mass change.
This approach relied on precise mass balance estimates from ICESat to train the model and thereby was able to remove the
effects of the changing scattering horizon in the radar data. This VC mass balance is given for monthly time steps (Simonsen
et al., 2021a), however the running-mean applied to derive radar elevation change will dampen the interannual variability of the
mass balance estimate from VC. This is especially true prior to 2010, after which the novel radar altimeter onboard CryoSat-2
allowed for a shortening of the data windowing from 5 to 3 years. This smoothing of the interannual variability is also seen in
the intercomparison between This Study and the VC MB, where in addition to the two end members of the time series (1992
and 2019) the years 1995, 1996, and 1998 seem to be outliers (Fig. 7). These years are notable for high MB which seems to be
captured less precisely by the older radar altimeters due to the longer temporal averaging.

## 6.5   IMBIE

The most widely cited estimate of Greenland mass balance today is the Ice-Sheet Mass Balance Inter-Comparison Exercise
2 (IMBIE2, The IMBIE Team (2019)). IMBIE2 seeks to provide a consensus estimate of monthly Greenland mass balance
between 1992 and 2018 that is derived from altimetry, gravimetry, and input-output ensemble members. There are two critical
methodological differences between This Study and IMBIE2. Firstly, the gravimetry members of IMBIE2 assess mass balance
of all Greenlandic land ice, including peripheral ice masses, while This Study only assesses mass balance of the ice sheet
proper. Secondly, the input-output members of IMBIE2 do not assess BMB, while This Study does.

The IMBIE2 composite record of ice-sheet mass balance equally weights three methods of assessing ice-sheet mass balance:
input-output, altimetry and gravimetry. Prior to c. 2003, however, IMBIE2 is derived solely from IO studies that explicitly
exclude BMB (MB is actually MB$^{*}$). After c. 2003, by comparison, IMBIE2 includes both satellite altimetry and gravimetry
records implicitly sample BMB. The representation of BMB in the composite IMBIE2 mass balance record therefore shifts
before and after c. 2003.

In comparison to mass balance assessed by IMBIE2, This Study has a small bias of ~ $-7$ Gt yr$^{-1}$ over the 26 calendar year
comparison period. This apparent agreement may be attributed to the compensating effects of IMBIE2 effectively sampling
peripheral ice masses and ignoring BMB, while This Study does the opposite and ignores peripheral ice masses but samples
BMB, equal to ~25 Gt yr$^{-1}$. Over the entire 26-year comparison period, the RMSE with IMBIE2 is $44$ Gt yr$^{-1}$ and the

correlation is `0.89`. This relatively high correlation highlights good agreement in interannual variability between studies, and the RMSE suggests that formal stated uncertainties of each study (c. ±30 to ±63 Gt yr$^{-1}$ for IMBIE2 and mean of 86 Gt yr$^{-1}$ for This Study) are indeed good estimates of the true uncertainty, as assessed by inter-study discrepancies.

## 7 Uncertainty

We treat the three inputs to the total mass balance (surface mass balance, discharge, and basal mass balance, or SMB, D, and BMB) as independent when calculating the total error. This is a simplification – the RCM SMB and the BMB$_{VHD}$ from RCM runoff are related, and D ice thickness and BMB$_{VHD}$ pressure gradients are related, and other terms may have dependencies. However, the two dominant IO terms, SMB inputs and D outputs, are independent on annual time scales, and for simplification we treat all terms as independent. We use Eq 3 and standard error propagation for SMB, D, and BMB terms (i.e., the square root of the sum of the squares of the SMB plus D plus BMB error terms). For D, extra work is done to calculate uncertainty between the last Mankoff et al. (2020b) D data (up to 30 days old, with error of ~9 % or ~45 Gt yr$^{-1}$) and the forecasted now-plus-7-day D (see Sect. 7.1). Table 3 provides a summary of the uncertainty for each input.

The final This Study MB uncertainty value shown in Table 3 comes from mean of the annual sum of the MB error term.

### 7.1 Discharge

The D uncertainty is discussed in detail in Mankoff et al. (2020b), but the main uncertainties come from unknown ice thickness, the assumption of no vertical shear at fast-flowing marine-terminating outlet glaciers, and ice density of 917 kg m$^{-3}$. Regional ice density can be significantly reduced by crevasses. For example, Mankoff et al. (2020c) identified a snow-covered crevasse field with 20 % crevasse density, meaning at that location regional firn density should be reduced by 20 %.

Temporally, D at daily resolution comes from ~12 day observations up-sampled to daily, and those ~12 day resolution observations come from longer time period observations (Solgaard et al., 2021). Because the velocity method uses feature tracking, it is correct on average but misses variability within each sample period (e.g., Greene et al. (2020)).

Spatially, discharge is estimated ~5 km upstream from the grounding lines for ice velocities as low as 100 m yr$^{-1}$. That ice accelerates toward the margin, but even ice flowing steady at 1 km yr$^{-1}$ would take 5 years before that mass is lost. However, at any given point in time, ice that had previously crossed the flux gate is calving or melting into the fjord. The discrepancy here between the flux gate estimated mass loss and the actual mass lost at the downstream terminus is only significant for glaciers that have had large velocity changes at some point in the recent past, large changes in ice thickness, or large changes in the location (retreat or advance) of the terminus. We do not consider SMB changes downstream of the flux gate, because the gates are temporally near the terminus for most of the ice that is fast-flowing, and the largest SMB uncertainty is at the ice sheet margin where there are both mask issues and high topographic variability.

The forecasted D uncertainty is the average historical uncertainty plus a 1 % increase per day for the past projected and forecasted period.

| Term | Uncertainty [±] | Notes |
|---|---|---|
| HIRHAM / HARMONIE SMB | 15 % | Langen et al. (2017). The mean accumulation bias (-5%) and ablation bias (-7%) tend to cancel out, but this cannot be expected to be the case on single-basin, short-term scales where uncertainty is estimated to be larger. |
| MAR SMB | 15 % | Fettweis et al. (2020). The mean bias between the model and the measurements was 15 % with a maximum of 1000 mmWE $yr^{-1}$. GrSMBMIP uses integrated values over several months of SMB, suggesting larger uncertainty of modeled runoff at the daily timescale. |
| RACMO SMB | 15 % | Noël et al. (2019). Average 5% runoff bias compared to annual cumulative discharge from the Watson River. Increases to a maximum of 20 % for extreme runoff years. |
| This Study SMB | 9 % | Average of 15 % SMB uncertainties above, assuming uncorrelated. |
| Reconstructed SMB | $\sim 20$ % | From Kjeldsen et al. (2015) Table 1. |
| Recent D | $\sim 45$ Gt $yr^{-1}$ | $\sim 9$ %. Mankoff et al. (2020b) updated (Mankoff, 2021). |
| Reconstructed D | $\sim 10$ % | From Kjeldsen et al. (2015) Table 1. |
| $BMB_{GF}$ | 50 % | 5.3 +4/-1.4 Gt $yr^{-1}$ from Karlsson et al. (2021) Table 1, using the average of the three available methods. |
| $BMB_{friction}$ | 20 % | 11.8 $\pm$3.4 Gt $yr^{-1}$ from Karlsson et al. (2021) Table 1. |
| $BMB_{VHD}$ | 15 % | MAR runoff uncertainty. |
| This Study MB | $\sim 86$ Gt $yr^{-1}$ | Eq 3, assuming all uncertainty is uncorrelated. |

**Table 3.** Summary of uncertainty estimates for products used in This Study. This is an approximate and simplified representation – RCM uncertainties are calculated separately for gain and loss terms, because SMB near 0 does not mean uncertainty is near 0. This is also why the final This Study uncertainty is presented with units [Gt $yr^{-1}$].

## 7.2 Regions of interest (ROI)

We work on the three different domains of the three RCMs, and expand the ROIs to match the RCMs (see Appendix E). However, some alignment issues cannot be solved. For example, we use BedMachine ice thickness to estimate $BMB_{VHD}$. Often, the largest $BMB_{VHD}$ occurs near the ice margin under ice with the steepest surface slopes. This is also where the largest runoff often occurs, because the ice margin, at the lowest elevations, is exposed to the warmest air. If these RCM ice grid cells with high runoff are anywhere inside the BedMachine ice domain, that runoff is still included in our $BMB_{VHD}$ estimates because it flows outward and passes through the BedMachine near-ice-edge grid cells with the large pressure gradients. However, any RCM ice runoff outside the BedMachine ice domain (ice thickness is 0) is ignored.

The MAR ice domain is 1,825,600 km$^2$ of which 1,708,400 km$^2$ (94 %) are covered by the BedMachine ice mask, and 26,400 km$^2$ (6 %) are not. This 6 % area contributes ~18 % of runoff on average (range of 16 % to 21 % from 2010 through 2019). This 18 % of runoff is excluded from the VHD calculations and likely contributes more than 18 % to the VHD term, because the border region of the ice sheet has the steepest gradients and the largest volume of subglacial flow.

We encourage RCM developers, BedMachine, and others to use a common and up-to-date mask (see Kjeldsen et al. (2020)).

## 7.3 Accumulating uncertainties

When accumulating errors as in Fig. 4, we use only the D and $BMB_{GF}$ uncertainty. The D uncertainty is primarily due to unknown ice thickness and is invariant in time, and the geothermal heat flux is steady state. SMB uncertainty is assumed to
515 have errors randomly distributed in time (for the purposes of Fig. 4). There may be time-invariant biases in the $BMB_{friction}$ and SMB fields, but treating all uncertainties as biases is incorrect - evidence for that comes from the six other MB estimates. This distinction between bias and random uncertainty is only done for Fig. 4 where errors accumulate in time. The provided data product contains one uncertainty field and does not distinguish between systematic and random uncertainty. We caution others in treating SMB uncertainty as random in time for analyses that go beyond the graphical display used here.

The shaded region in Fig. 4 representing the uncertainty for This Study is computed as a 365 day rolling smooth from 1840 through 1999 of the above-described uncertainty, 1/365th of the annual error at now + 7 days, and a linear blend, from 2000 to now + 7 days, between the smoothed reconstructed uncertainty and the present and future more variable uncertainty.

The Mouginot et al. (2019), Colgan et al. (2019), and Kjeldsen et al. (2015) products all provide an error estimate, but do not distinguish between temporally fixed errors (biases; should accumulate in time) vs. temporally random errors.

We treat the Mouginot et al. (2019) data the same as This Study. Discharge uncertainty is treated as a bias and accumulates, and surface mass balance uncertainty is treated as random and does not accumulate.

The Colgan et al. (2019) vs. this study bias and RMSE are -32 and 59 Gt yr$^{-1}$ respectively. This suggests that in any given year, there could be up to -32 ± 59 or +27/-91 Gt yr$^{-1}$ departure from This Study. From this, we assign a 32 Gt yr$^{-1}$ bias (35 %; accumulates in time) and a 59 Gt yr$^{-1}$ RMSE (65 %; random in time).

The adjusted Kjeldsen et al. (2015) data have 0 surface mass balance and discharge bias by definition (Sect. 5.4), but Fig. 4 displays the unadjusted data, and we apply a 36 Gt yr$^{-1}$ accumulating uncertainty from the unadjusted D bias (Fig. 3).

## 7.4 Peripheral ice masses

Greenland's peripheral glaciers and ice caps are not included in this product. Nonetheless, we briefly summarize recent mass balance estimates of these areas. Greenlandic peripheral ice contributes more runoff per unit area than the main ice sheet –
535 they are < 5 % of the total ice area but contribute ~15 to 20 % of the whole island mass loss (Bolch et al., 2013). From 2003 to 2009 and using the VC method (altimetry), Gardner et al. (2013) estimate -38 ±7 Gt yr$^{-1}$ peripheral mass balance. From 2006 to 2016 and using the VC method (DEM differencing), Zemp et al. (2020) estimate -51 ±17 Gt yr$^{-1}$ peripheral mass balance, using Rastner et al. (2012) delineations.

## 8  Results

From the 181 complete years of data (excluding partial 2021), the mean mass balance is -77 ±125 Gt yr$^{-1}$, with a minimum of -428 ±110 Gt in 2012 (SMB of 87 ±8 Gt, D of 485 ±46 Gt, BMB of 29 ±6 Gt) and a maximum of 142 ±83 Gt yr$^{-1}$ Gt in 1996 (SMB of 584 ±53 Gt, D of 420 ±39 Gt, BMB of 21 ±5 Gt).

At the decadal average, the following trends are apparent. Surface mass balance has decreased from a high of ~450 Gt yr$^{-1}$ in the 1860s to low of ~260 Gt yr$^{-1}$ in the 2010s. SMB variability has also increased during this time. Discharge has increased 545 slightly from a low of ~375 Gt yr$^{-1}$ in the 1860s to a high of ~490 Gt yr$^{-1}$ in the 2010s. Basal mass balance, from runoff as a proxy, had a high of 26 ±16 Gt yr$^{-1}$ in the 1930s and a low of 22 ±5 Gt yr$^{-1}$ in the 1990s, but as with runoff, is increasing in recent decades.

The total mass balance decadal trend from the 1840s through the 2010s is one of general mass decrease and increased intra-decadal variability. The record begins in the 1840s with ~-10 Gt yr$^{-1}$, has only one (of 19) decades with a mass gain (~50 Gt 550 yr$^{-1}$ in the 1860s), and a record low of ~-250 Gt yr$^{-1}$ in the 2010s.

## 9  Data availability

The RCM surface mass balance, and the VHD basal mass balance components are updated daily, the discharge approximately every 12 days, and all are used to produce the final daily-updating product. The data area available at https://doi.org/10.22008/FK2/OHI23Z (Mankoff et al., 2021), with all historical (daily updated) versions archived.

As part of our commitment to make continual and improving updates to the data product, we introduce a GitHub database (https://github.com/GEUS-Glaciology-and-Climate/mass_balance/; last visited October 4, 2021) where users can track progress, make suggestions, discuss, report and respond to issues that arise during use of this product.

## 10  Conclusions

This study is the first to provide a dataset containing more than a century and real time estimates detailing the state of Greenland 560 ice sheet mass balance, with regional or sector spatial and daily temporal resolution products of surface mass balance, discharge, basal mass balance, and the total mass balance.

IMBIE2 highlights that during the GRACE satellite gravimetry era (2003 through 2017), there are usually more than twenty independent estimates of annual Greenland ice sheet mass balance. Just two independent estimates, however, are available prior to 2003. This study will therefore provide additional insight on ice sheet mass balance during the late 1980s and 1990s. 565 IMBIE2 also highlights how the availability of mass balance estimates declines in the year prior to IMBIE2 publication. This reflects a lag period during which mass balance assessments from non-operational products are undergoing peer-review. The operational nature of this product supports the timely inclusion of annual MB estimates in community consensus reports such as those from IMBIE and the IPCC.

As such, the data products provided in this study present the first operational monitoring of the Greenland ice sheet total mass

balance and its components. One property of the input-output approach used in This Study is the explanatory capabilities of the data products, allowing scrutiny of the physical origins of recorded mass changes. By excluding peripheral ice masses, this study allows and invites anyone to keep an eye on the current evolution of the Greenland ice sheet proper. However, as the spatial resolution of RCMs increase and estimates of peripheral ice thickness become available, our setup allows inclusion of these ice masses to generate a full Greenland-wide product. Moreover, as the determination of each of the individual components of the

ice sheet mass balance is expected to improve over time through international research efforts, the total mass balance product presented will also be able to improve, as it is sustained by the Danish-Greenlandic governmental long-term monitoring effort – the Programme for Monitoring of the Greenland ice sheet (PROMICE).

# Appendix A: RCM differences

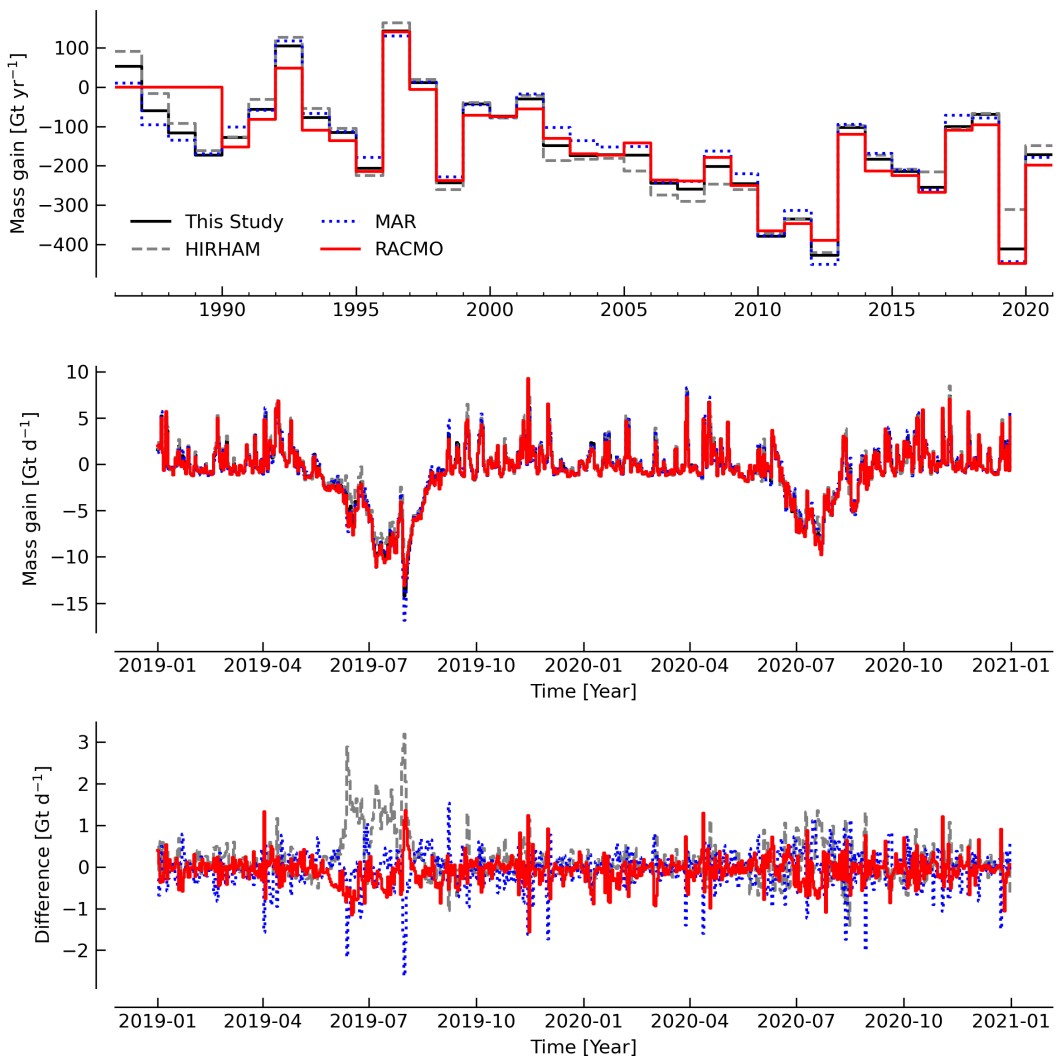

**Figure A1.** Comparison of This Study combined RCM product and the HIRHAM/HARMONIE, MAR, and RACMO RCMs. Results shown here are MB, not SMB, but the same D and BMB have been subtracted from each SMB product. Top panel: annual MB for entire time series. Middle panel: Example two years (2019 and 2020) at daily resolution. Bottom panel: Difference between the three RCM MB products and This Study RCM-averaged product, for the same data shown in the middle panel.

**Appendix B:  Mouginot 2019 by region**

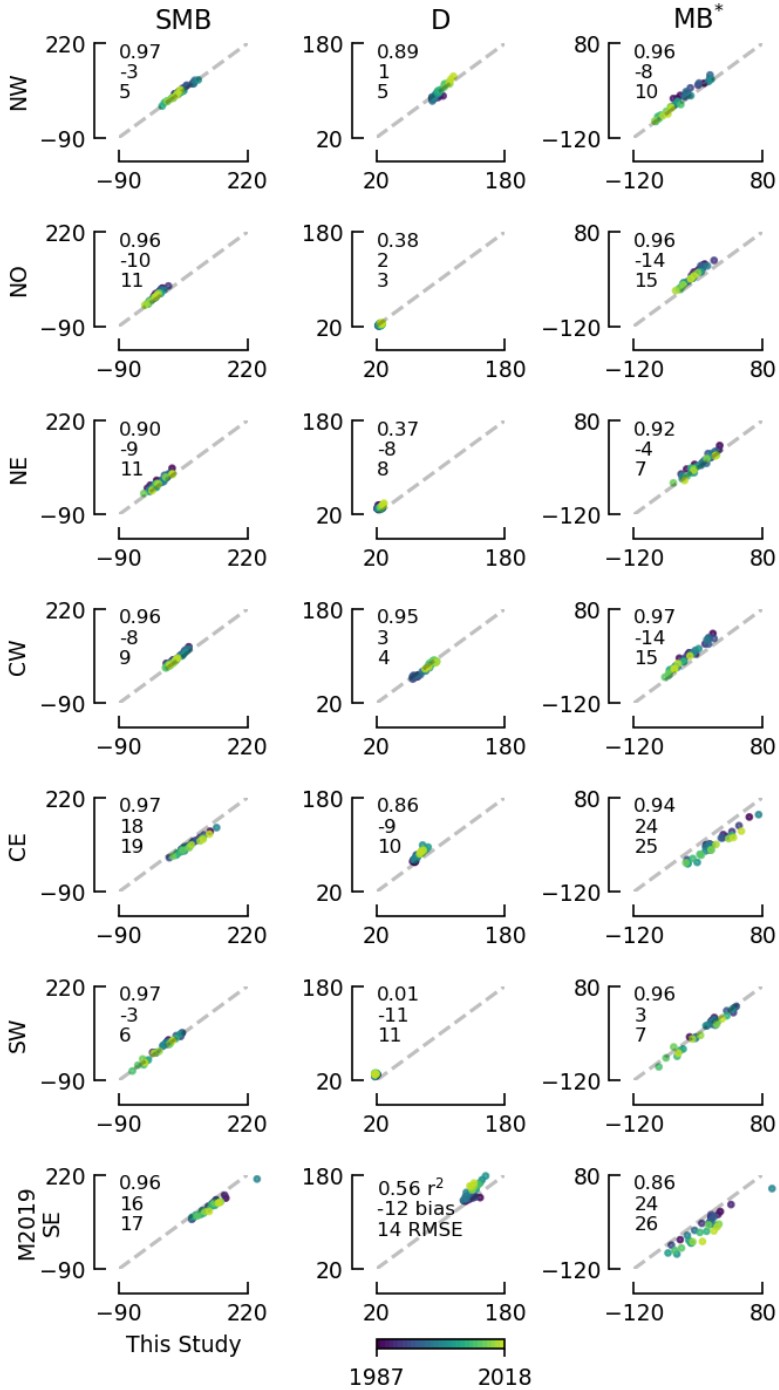

**Figure B1.** Comparison between This Study (excluding BMB) and Mouginot et al. (2019). Same data and display as Fig. 5 except here displayed by Mouginot and Rignot (2019) region. Numbers in each graph show $r^2$, bias, and RMSE, from top to bottom, respectively. All axes units are Gt yr$^{-1}$. Plotted numbers represent the last two digits of the year.

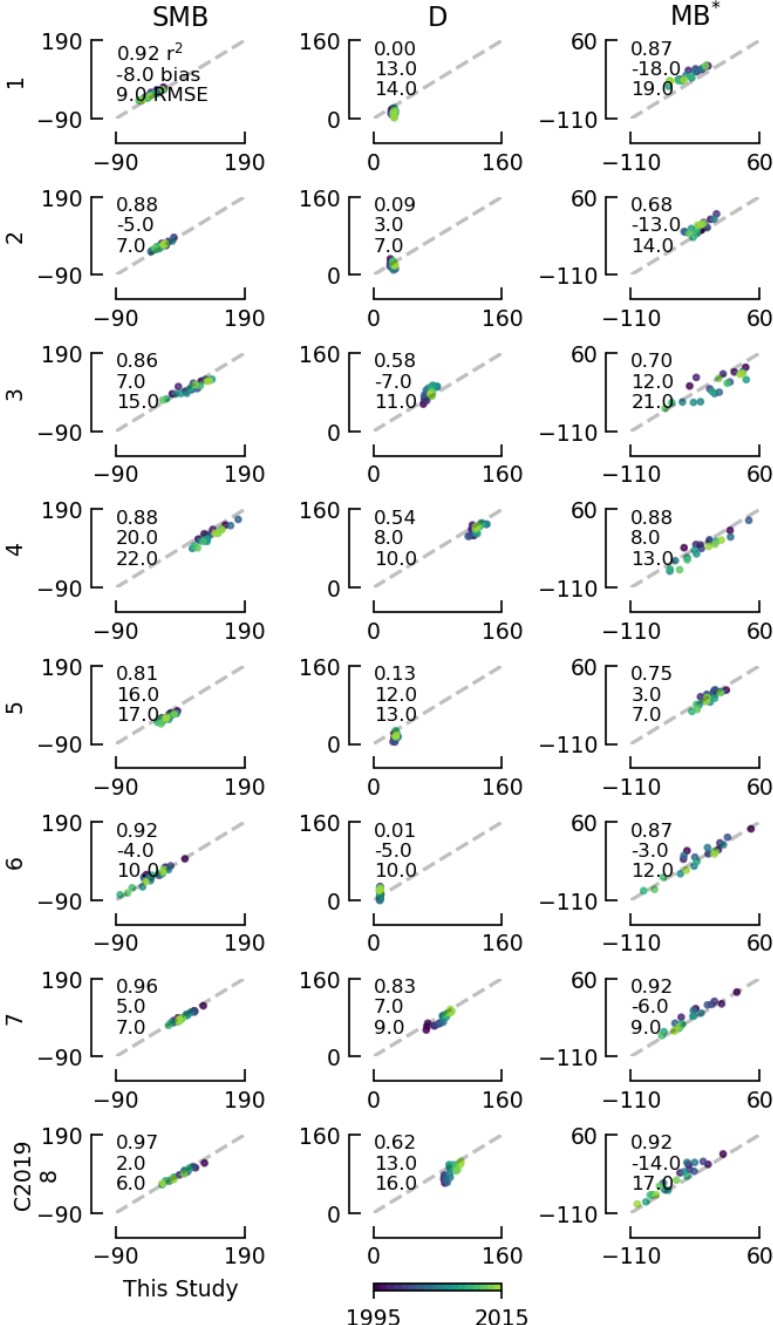

**Figure C1.** Comparison between This Study (excluding BMB) and Colgan et al. (2019). Same data and display as Fig. 6 except here displayed by Zwally et al. (2012) sector. Numbers in each graph show $r^2$, bias, and RMSE, from top to bottom, respectively. All axes units are Gt yr$^{-1}$. Plotted numbers represent the last two digits of the year.

**Appendix D: Reconstructed runoff**

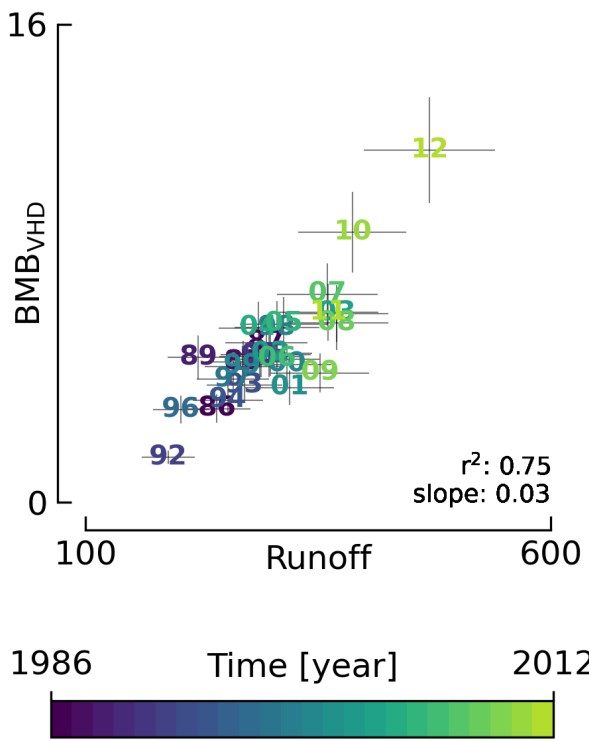

**Figure D1.** Comparison between MAR runoff and basal viscous heat dissipation derived from that runoff. The slope is used to estimate the reconstructed $BMB_{VHD}$ from reconstructed runoff (see Sect. 5.3). Axes units are Gt yr$^{-1}$. Plotted numbers represent the last two digits of the year.

## Appendix E:  RCM coverage

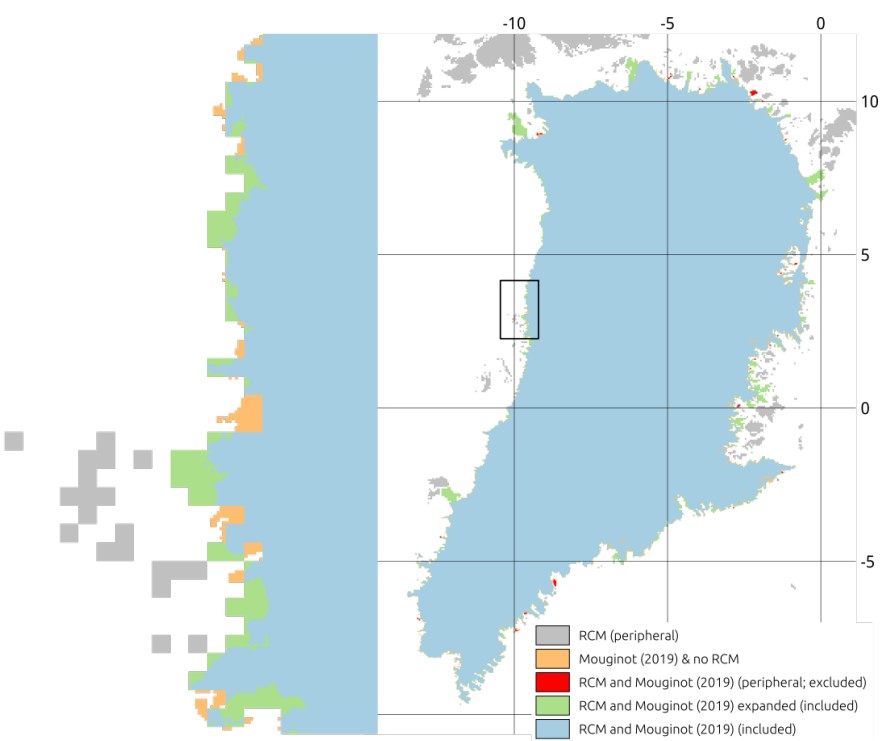

**Figure E1.** HIRHAM RCM coverage by Mouginot and Rignot (2019). Coverage of HIRHAM by Zwally et al. (2012), and MAR and RACMO by Mouginot and Rignot (2019) and Zwally et al. (2012) is similar to graphic shown here (See section 5.5 for discussion of RACMO coverage issues). HIRHAM latitude and longitude covers the equator because we work on the native HIRHAM rotated pole coordinate system.

## Appendix F:  Software

This work was performed using only open-source software, primarily `GRASS GIS` (Neteler et al., 2012), CDO (Schulzweida, 2019), NCO (Zender, 2008), GDAL (GDAL/OGR contributors, 2020), and `Python` (Van Rossum and Drake Jr, 1995), in particular the `Jupyter` (Kluyver et al., 2016), `dask` (Dask Development Team, 2016; Rocklin, 2015), `pandas` (McKinney, 2010), `geopandas` (Jordahl et al., 2020), `numpy` (Oliphant, 2006), `x-array` (Hoyer and Hamman, 2017), and `Matplotlib` (Hunter, 2007) packages. The entire work was performed in `Emacs` (Stallman, 1981) using `Org Mode` (Schulte et al., 2012) on GNU/Linux and using many GNU utilities. The `parallel` (Tange, 2011) tool was used to speed up processing.

## Appendix G: CRediT

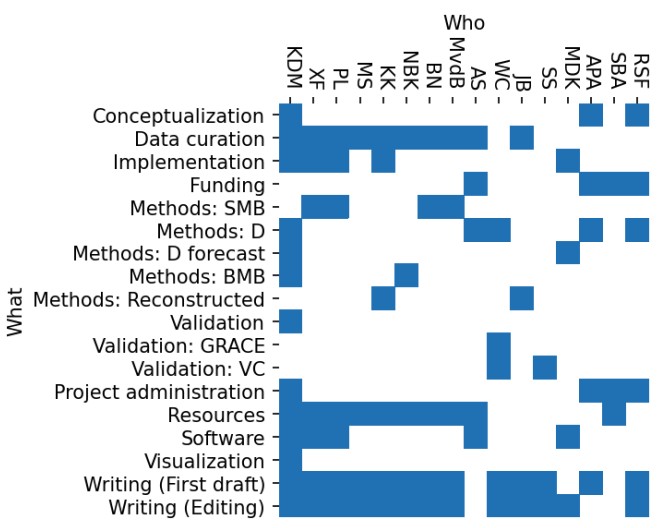

**Figure G1.** Author contributions following the CRediT system (Allen et al., 2014; Brand et al., 2015; Allen et al., 2019)

*Author contributions.* {

Author contribution is captured following the CRediT system (Allen et al., 2014; Brand et al., 2015; Allen et al., 2019) and shown graphically in Figure G1. The following authors contributed in the following ways. Conceptualization: KDM, APA, and RSF. Curation: KDM, XF, PL, MS, KK, NBK, BN, MvdB, AS, and JB. Implementation: KDM, XF, PL, KK, and MKD. Funding: AS, APA, SBA, and RSF. SMB methods: XF, PL, BN, and MvdB. D methods: KDM, WC, AS, MKD, APA, and RSF. BMB methods: NBK and KDM. Validation (general): KDM. Validation GRACE: WC. Validation VC: WC and SS. Reconstruction methods: KK, JB, and KDM. Project admin: KDM, APA, SBA, and RSF. Resources: KDM, XF, PL, MS, KK, NBK, BN, MvdB, AS, and SBA. Software: KDM, XF, PL, AS, and MKD. Visualization: KDM. Writing: KDM, XF, PL, MS, KK, NBK, BN, MvdB, WC, JB, SS, APA, and RSF.}

*Competing interests.* The authors declare that they have no conflict of interest.

*Acknowledgements.* The editor, and anonymous reviewers (Anonymous, 2021a, b) provided feedback and helped improve this paper.

Financial Support: Funding was provided by the Programme for Monitoring of the Greenland ice sheet (PROMICE). Parts of this work were funded by the INTAROS project under the European Union's Horizon 2020 research and innovation program under grant agreement no.

727890. B. Noël was funded by the NWO VENI grant VI.Veni.192.019. MvdB acknowledges support from the Netherlands Earth System Science Centre (NESSC).

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
