# Peer review of "Greenland ice sheet mass balance from 1840 through next week"

_Earth System Science Data, 2021_

## Referee Comment (RC1)

**Review of Mankoff et al. (2021)**

General comments:

This study describes a new operational mass balance product of the Greenland Ice Sheet produced using the input-output method. The different external products used in this study, and the methods for the derivation of this new mass balance estimate are sound and well described. This new mass balance estimate is also compared against independent estimates of mass balance, showing overall good agreement.

I have a few general comments that should not be hard to address:
- In Section 4, I recommend adding a table to summarise the different data products used to generate the final mass balance estimate with the different periods they cover and what field(s) they provide
- I think that the consistency of the description of the three RCMs in Section 4.3. can be improved by matching the level of details between the different SMB models/products described (for instance what ice mask is used in each RCM)
- I'm not convinced about the difference in MB* between the various IO estimates (Mouginot/This Study and Colgan/This Study) being attributed to differences in SMB as described in the text in sections 6.1 and 6.2. From the figures and the SMB/MMB comparisons, it seems that the differences between the various IO estimates arise from differences in MMB rather than SMB.

Overall, this is a great effort to generate the first operational product of Greenland mass balance by making use of already existing methods and data products, which I think will be of interest to anyone interested in the state of the Greenland Ice Sheet. Therefore, I'm happy to recommend this paper to be published after minor revisions.

I made some specific comments and suggestions below, which I hope will help improve this paper.

Specific comments:

**L1:** 'Greenland Ice Sheet (GIS)'

**L5-6:** please rephrase this sentence to distinguish the products (from which mass loss is estimated) and the processes (from which the mass loss originates)

**L10:** 'general agreement with  six other products'

**L19:** 'processes'

**L18-21:** add typical spatial resolution of the GMB and VC estimates and the typical temporal resolution of VC estimates to quantify what is lower/higher

**L25-26:** can you be a bit more specific here: state how frequent ice velocity updates are and the reason for this new capability (for instance new satellite missions (e.g. Sentinel) allowing better temporal sampling of ice velocity time-series)

**L25:** I would also add that IO is limited by the scarcity of thickness data in some areas

**L31-32:** be more specific here and state the positive and negative SMB fields

**L35:** define 'marine mass balance' here

**L35:** I'm not sure 'forecasted' is the right word here, as rather than being forecasted the MMB is updated assuming steady-state conditions since the last velocity update

**L61:** what 'properties'?

**L61-62:** 'These  Greenland sum data'

**L67:** Not sure what you mean by 'includes the restricted data to 1840' maybe rephrase with something along the lines 'The ice-sheet-wide product includes data from 1840 through next week while the sector and region-scale products includes data from 1986 through next week.'

**L70:** 'are external to this work'.

**L75:** By curiosity, why do you use runoff from MAR only? I assume that runoff is also modelled by HIRHAM and RACMO and that the same approach used for the SMB models (i.e. combining the models when they overlap and using MAR runoff from yesterday through next week) could be applied for runoff?

**L97:** specify the min/max of the layer thicknesses

**L106-107:** can you add a few words to specify the benefits of including these observations? Does the model perform better?

**L113:** specify the min/max thickness of the firn layers in MAR

**L110-123:** It's fine to refer the reader to Fettweis et al. (2020) but it would be good to expand this paragraph a bit more, and ensure that the same level of details is provided in each RCM description

**L114:** add citation for the NCEP-NCARv1 reanalysis

**L116:** 'GridMARv3.10', is it different from MARv3.10?

**L116:** no need to mention the inclusion of a new module if it is not used here

**L121:** please provide more background on the recent SMB decrease and the validation with GRACE. This is important as both the SMB and runoff fields are used in this study

**L122:** 'increase  in runoff'

**L129-130:** is runoff also forecasted? Also add that you use SMB, forecasted SMB and runoff

**L165:** replace 'add [..] a later end date' by 'extend the reconstruction in time up to the end of 2012'

**L182-183:** 'is  updated'

**L186:** which surface elevation change product do you use?

**L211-212:** what proportion of the Greenland Ice Sheet bed area is frozen, uncertain or thawed? Can you add the % bed area covered in the text?

**L212:** ', '

**L215:**  at the ice-sheet scale

**L226:** 'these products are the most  recent'

**L261:**  BMB$_{VHD}$

**L272:** missing word 'We compute $h$ and [?] from that streams and outlets'

**L307-308:** can you say how much smaller are the Zwally sectors and Mouginot regions compared to the RCM ice domains and what proportion of the SMB losses is not considered if the RCM domain is cropped to the Zwally/Mouginot delineations?

**Table 1:** For IMBIE2, BMB is included in both GMB and VC

**Table 1:** The numbers in the table and in the related figures are slightly different, especially for the bias values (for instance for Mouignot the bias is -4 Gt/yr based on Table 1 and -2 Gt/yr based on Figure 5).

**L339:** Isn't the MB* difference between This Study and Mouginot dominated by the MMB term rather than the SMB term, with the SMB disagreement adding only a small amount of noise? (You showed before a 23 Gt yr$^{-1}$ bias in MMB and a very good agreement in terms of SMB between This study and Mouginot)

**L363-365:** Here as well, it seems that the difference in MB* between This Study and Colgan is dominated by differences in MMB, rather than differences in SMB? Or do you speak in terms of short-term temporal variability when you're referring to 'the variation in Colgan et al (2019) MB*'? Either way, this needs to be clarified.

**L368:** 'peripheral glaciers and ice caps'

**L379:** ice sheet boundaries

**L391-393:** Could the disagreement between This Study and the VC estimates in 1992 and 2019 come from the fact that these two years are the end members of the time-series (perhaps edge effects could be the origin of the disagreement), rather than being driven by changes in the radar scattering horizon? 2012 was also an extreme melt year with the scattering horizon of the radar being shifted upwards closer to the ice sheet surface; however there is a good agreement between This Study and the VC estimates in this particular year (Figure 7b).

**L409:** 'assesses'

**L449:** 'and these grid cells are ignored. It is ignored.'

**Figure 5 and Figure B1:** on Figure 5 there is a very good agreement in SMB between This Study and Mouginot (r$^2$ 0.97 and bias -1) but Figure B1 suggests otherwise. It seems from Figure B1 that the agreement in SMB is much lower across all the regions of the Greenland Ice Sheet with r$^2$ <0.40 for all regions?

**Figures B1 and C1:**

- Is the BMB term included in the MMB and MB* terms here? Specify this in the caption.
- Maybe use a different yrange for the y-axis for the different regions/sectors as it is difficult to read the figures in some cases (for instance NO, NE or SW MMB)

---

## Author Comment (AC1)

**Reply to Reviewers**

**Ken Mankoff *et al.**

Comments from reviewers are in normal font and differentiated from **the replies that use a bold colored font** .

**Contents**

**1 Reviewer 1**

**1.1 General comments**

Review of Mankoff et al. (2021)

This study describes a new operational mass balance product of the Greenland Ice Sheet produced using the input-output method. The different external products used in this study, and the methods for the derivation of this new mass balance estimate are sound and well described. This new mass balance estimate is also compared against independent estimates of mass balance, showing overall good agreement.

I have a few general comments that should not be hard to address:

- In Section 4, I recommend adding a table to summarise the different data products used to generate the final mass balance estimate with the different periods they cover and what field(s) they provide

**Added.**

- I think that the consistency of the description of the three RCMs in Section 4.3. can be improved by matching the level of details between the different SMB models/products described (for instance what ice mask is used in each RCM)

**We've attempted to homogenize the RCM sections.**

- I'm not convinced about the difference in MB* between the various IO estimates (Mouginot/This Study and Colgan/This Study) being attributed to differences in SMB as described in the text in sections 6.1 and 6.2. From the figures and the SMB/MMB comparisons, it seems that the differences between the various IO estimates arise from differences in MMB rather than SMB.

**We have re-phrased this and clarified that the difference is due to MMB, but it is not apparent because the interannual variability is dominated by SMB.**

Overall, this is a great effort to generate the first operational product of Greenland mass balance by making use of already existing methods and data products, which I think will be of interest to anyone interested in the state of the Greenland Ice Sheet. Therefore, I'm happy to recommend this paper to be published after minor revisions.

**We're happy you like the concept and the implementation.**

I made some specific comments and suggestions below, which I hope will help improve this paper.

**We respond to all of your comments below.**

**1.2 Specific comments**

L1: 'Greenland Ice Sheet (GIS) '

**We removed all abbreviations for GIS and GrIS.**

L5-6: please rephrase this sentence to distinguish the products (from which mass loss is estimated) and the processes (from which the mass loss originates)

**We have rephrased the sentence.**

L10: 'general agreement  among six other products'

**We choose to keep among.**

1. **Mixed or mingled; surrounded by.**
2. **Conjoined, or associated with, or making part of the number of; in the number or class of.**
3. **Expressing a relation of dispersion, distribution, etc.; also, a relation of reciprocal action.**

**Also because I read a grammar book that states "between" is for 2 options and "among" is for 3 or more. If the proof editors prefer 'with' or 'between' I'll change it then.**

L19: 'processes es '

**process → processes**

L18-21: add typical spatial resolution of the GMB and VC estimates and the typical temporal resolution of VC estimates to quantify what is lower/higher

**Done.**

L25-26: can you be a bit more specific here: state how frequent ice velocity updates are and the reason for this new capability (for instance new satellite missions (e.g. Sentinel) allowing better temporal sampling of ice velocity time-series)

**Done.**

L25: I would also add that IO is limited by the scarcity of thickness data in some areas

**Done.**

L31-32: be more specific here and state the positive and negative SMB fields

**Clarified that we use SMB as an input, and do not access the individual components. We list the positive and negative components anyway.**

L35: define 'marine mass balance' here

**Done.**

L35: I'm not sure 'forecasted' is the right word here, as rather than being forecasted the MMB is updated assuming steady-state conditions since the last velocity update

**You are correct we did not 'forecast'. However, we've improved the methods here and now 'forecasted' and 'projected' are appropriate.**

L61: what 'properties'?

**Re-written.**

L61-62: 'These  Greenland sum data'

**Done**

L67: Not sure what you mean by 'includes the restricted [sic] data to 1840' maybe rephrase with something along the lines 'The ice-sheet-wide product includes data

from 1840 through next week while the sector and region-scale products includes data from 1986 through next week.'

**Rephrased.**

L70: ' are external to this work'.

**Added.**

L75: By curiosity, why do you use runoff from MAR only? I assume that runoff is also modelled by HIRHAM and RACMO and that the same approach used for the SMB models (i.e. combining the models when they overlap and using MAR runoff from yesterday through next week) could be applied for runoff?

**We opted to use only MAR runoff because MAR data included runoff in the initial delivery, while RACMO and HIRHAM included only the SMB field to save transfer bandwidth and storage space. While using all three may be more robust and in-line with the rest of the work, this term is small and the result would not change by including an ensemble BMB$_{VHD}$ term. Using only one RCM saves development and compute time and effort.**

L97: specify the min/max of the layer thicknesses

**Specified.**

L106-107: can you add a few words to specify the benefits of including these observations? Does the model perform better?

**Citation added.**

L113: specify the min/max thickness of the firn layers in MAR

**Done.**

L110-123: It's fine to refer the reader to Fettweis et al. (2020) but it would be good to expand this paragraph a bit more, and ensure that the same level of details is provided in each RCM description

**We have expanded the MAR section so it contains a similar level of detail as the HIRHAM and RACMO sections.**

L114: add citation for the NCEP-NCARv1 reanalysis

**Re-written.**

L116: 'GridMARv3.10', is it different from MARv3.10?

**Rewritten.**

L116: no need to mention the inclusion of a new module if it is not used here

**Removed.**

L121: please provide more background on the recent SMB decrease and the validation with GRACE. This is important as both the SMB and runoff fields are used in this study

**Citation added.**

L122: 'increase  in runoff'

**Changed.**

L129-130: is runoff also forecasted? Also add that you use SMB, forecasted SMB and runoff

**Runoff was not but is now forecasted. Text revised.**

L165: replace 'add [..] a later end date' by 'extend the reconstruction in time up to the end of 2012'

**Changed.**

L182-183: 'is  updat  ed'

**Changed.**

L186: which surface elevation change product do you use?

**Citation added.**

L211-212: what proportion of the Greenland Ice Sheet bed area is frozen, uncertain or thawed? Can you add the % bed area covered in the text?

**Done.**

L212: ',  respectively  '

**Removed parentheses**

L215:  at the ice-sheet scales

**Changed 'on' to 'at'**

L226: 'these products are the  and recent'

**Removed.**

L261:  BMB$_{VHD}$

**Removed.**

L272: missing word 'We compute h and [?] from that streams and outlets'

**The "that" refers to the $h$. $h$ is the only input needed for flow routing. We now explicitly state, "We compute $h$ and from $h$ streams and outlets".**

L307-308: can you say how much smaller are the Zwally sectors and Mouginot regions compared to the RCM ice domains and what proportion of the SMB losses is not considered if the RCM domain is cropped to the Zwally/Mouginot delineations?

**Text updated both to quantify the missing area in Mouginot and Zwally, and the missing SMB from that area. Text also updated to quantify the VHD losses from BedMachine vs. MAR alignment.**

Table 1: For IMBIE2, BMB is included in both GMB and VC

**Added "or VC"**

Table 1: The numbers in the table and in the related figures are slightly different, especially for the bias values (for instance for Mouignot the bias is -4 Gt/yr based on Table 1 and -2 Gt/yr based on Figure 5).

**Fixed.**

L339: Isn't the MB* difference between This Study and Mouginot dominated by the MMB term rather than the SMB term, with the SMB disagreement adding only a small amount of noise? (You showed before a 23 Gt yr$^{-1}$ bias in MMB and a very good agreement in terms of SMB between This study and Mouginot)

**You are correct. Clarified.**

L363-365: Here as well, it seems that the difference in MB* between This Study and Colgan is dominated by differences in MMB, rather than differences in SMB? Or do you speak in terms of short-term temporal variability when you're referring to 'the variation in Colgan et al (2019) MB*'? Either way, this needs to be clarified.

**Correct & clarified.**

L368: 'peripheral glaciers and ice caps '

**Clarified inclusion of peripheral ice masses.**

L379: ice sheet boundar ies

**Changed.**

L391-393: Could the disagreement between This Study and the VC estimates in 1992 and 2019 come from the fact that these two years are the end members of the time-series (perhaps edge effects could be the origin of the disagreement), rather than being driven by changes in the radar scattering horizon? 2012 was also an extreme melt year with the scattering horizon of the radar being shifted upwards closer to the ice sheet surface; however there is a good agreement between This Study and the VC estimates in this particular year (Figure 7b).

**It could be edge-effects, although many other years (1995, 1998, 1996) are of similar disagreement. We have added a comment on possible edge effects.**

L409: 'assess +es+'

**Removed 'es'**

L449: ' and these grid cells are ignored. It is ignored. '

**Changed.**

Figure 5 and Figure B1: on Figure 5 there is a very good agreement in SMB between This Study and Mouginot (r$^2$ 0.97 and bias -1) but Figure B1 suggests otherwise. It seems from Figure B1 that the agreement in SMB is much lower across all the regions of the Greenland Ice Sheet with r2 <0.40 for all regions?

**Thank you for catching that. The Mouginot data has dates centered in the year, and we correctly aligned dates for the GIS-summed graphic in the main section, but had not done this correctly in the Appendix. The agreement between This**

**Study and Mouginot (2019) is now also apparent at the region-level shown in the Appendix.**

Figures B1 and C1: Is the BMB term included in the MMB and MB* terms here? Specify this in the caption.

**By definition BMB is not included because MB$^{*}$ is used. We now add "(excluding BMB)" to the figure captions to make this explicit.**

Figures B1 and C1: Maybe use a different yrange for the y-axis for the different regions/sectors as it is difficult to read the figures in some cases (for instance NO, NE or SW MMB)

**We made earlier versions with each sector or region y-axis scaled, but opted for this display instead. Yes it makes it difficult to read individual year data for some cases as you pointed out, but highlights inter-sector or inter-region differences (for example, the NO, NE, and SW MMB you highlighted clearly contribute minimal MMB, while SE contributes the most). We prefer to keep the current display which highlights different properties of the data.**

**2 Review 2 (Andrew Shepherd)**

This paper describes an updated mass budget (input output method) for the Greenland Ice Sheet spanning the period 1840 to the present day. From 1986 onwards, the mass budget components are derived from 3 regional climate models, satellite-based estimates of ice discharge, and estimates of basal melting. Prior to 1986, all components are derived from a reconstruction of the surface mass balance and runoff. It is a valuable dataset and deserves to be published.

**2.1 Main concerns**

I have the following 6 main concerns.

This study introduces a new term – marine mass balance, which is the sum of solid ice discharge and submarine ice melting. I am not convinced that the two terms should be added in this way as one is a lateral flux of ice and the other is a vertical flux of water, and they are in any case representative of different process. It seems to me that they should be kept separate.

**In Greenland, unlike Antarctica, submarine melt is usually lateral, not vertical, assuming Greenland marine terminating glaciers have vertical ice fronts, while Antarctic ice shelves float with mostly horizontal surfaces in the water. We disagree that 'melting' is a flux of water - or at least only water. 'Melting' also requires ice. Regardless of the medium both are mass volume flow rate units, not flux units.**

**I'm not aware that it is possible to separate these terms. Their different values are not known at most glaciers and at most times. There have been a few studies that have examined the ratio of calving to melting (e.g Enderlin et al. (2013) and De Andrés et al. (2018)), or estimate melting from runoff (e.g. Slater et al. (2019) and De Andrés et al. (2021)) but they are poorly constrained and not operational products.**

**The term is new, but not the data. We point out in the text (L177 of reviewed document) that this is the 'discharge' data (and term) already used in other papers (e.g. Mankoff et al. (2020), King et al. (2020), King et al. (2018), and Enderlin et al. (2014)). We use 'marine mass balance' rather than 'discharge' because a) it matches the other terms (surface mass balance and basal mass balance) and b) we think it is important to highlight that what may be 'solid ice discharge' across the flux gate, splits into calving or melting when it enters a fjord. This distinction is critical for studies of the down-fjord freshwater budget, because submarine melting introduces freshwater at the terminus, and calving introduces freshwater elsewhere, where the icebergs melt.**

The submarine melting term is required because the flux gate is positioned in places over floating ice. I question why this is done, given that it adds complexity and uncertainty to the result, and it also means that the result is no longer pertaining to the grounded ice sheet. My recollection is that this term was erroneously included in one mass budget assessment for the Antarctic ice sheet and has since been removed (Rignot et al., 2011). Why not position the flux gate so that no submarine melting is required? I would also like to see more discussion on this submarine melting term and some presentation of the data to illustrate its magnitude and spatial distribution.

The flux gates are not over floating ice. The flux gates are ~5 km upstream from all grounding lines (L179 of reviewed document). We do not know the location and magnitude of submarine melt. This is an area of active research. As soon as this data becomes available we plan to incorporate it into the continually updating product from Mankoff et al. (2020) that is used as an input to This Study.

We could then separate $MMB_{calving}$ from $MMB_{melting}$, but even then $MMB_{calving}$ should be split into big bergs and small slumping events that effectively melt immediately. Also, in order to be consistent we should then also separate SMB gain (snowfall, rainfall, condensation) from SMB loss (melt, evaporation, sublimation), etc. All this suggests a major update, perhaps as Edition #2 of this paper and product.

It is acknowledged that there are inconsistencies in the domains of the various input datasets at the ice sheet margin, and that this is a location where large signals are present. Some effort is made to align the dataset, but I could not follow the explanation in full. In any case, I would like to see some exploration of the sensitivity of the results to the location of the boundary to be sure that it is not erroneously positioned. This is somehow related to the need to include a submarine melting term; at the location between grounded and floating ice there are typically very large variations in surface and basal ice melting and I am not convinced they can be reliably separated. It would be interesting to see how the various mass balance terms vary as a function of domain size while for example eroding the domain by a pixel at a time.

This is not related to the submarine melt term. That occurs downstream of the MMB flux gates.

If we introduced domain errors by cropping or aligning things we agree that a sensitivity study should be performed. However, we do not introduce domain errors. Your statement that "some effort is made to align the dataset" does not capture the amount of effort we put into correct alignment. We specifically work on the native RCM domains (including rotated pole for the HIRHAM/HARMONIE domain) to make sure that all RCM inputs and outputs are included in this work. We refer readers to Kjeldsen et al. (2020) for an exploration domain issues.

We now include a graphic in the Appendix showing the RCM domain, the Zwally et al. (2012) and Mouginot et al. (2019) (un-expanded), and the expanded Zwally et al. (2012) and Mouginot et al. (2019) that entirely cover the RCM domains with zero losses. We have added paragraphs quantifying the various overlaps.

The one case where we cannot align things correctly is MAR runoff used to estimate $BMB_{VHD}$. That term is one of three that make up ~24 Gt yr$^{-1}$ additional mass loss. Because it has a strong seasonal cycle one could argue that the error is 0 Gt yr$^{-1}$ of the BMB term during the winter when there is no runoff, and more during the summer when there is significant runoff. We now quantify the mask area misalignment and 'lost' runoff for that term.

Prior to 1986, the ice sheet mass balance is derived form a scaled reconstruction of surface mass balance and an empirical model of the ice discharge. The SMB is scaled to fit the observations derived from regional climate models post 1986. The ice discharge is modelled as a linear function of the reconstructed runoff based on

a fit to data recorded between 2000 and 2012. This interval is widely recognized to be a highly anomalous period in the Greenland Ice Sheet's history (see e.g. Boers & Rypdale, 2021), and I question the validity of extrapolating a relationship between runoff and discharge obtained from this period to other times. While I recognize that the method was introduced by other authors in 2013, this was prior to the anomaly was established and I believe the validity of the approach needs to be revisited and reestablished. What evidence is there that runoff and discharge were correlated prior to 2000? What happens if a different period is chosen?

**The method for scaling discharge from runoff was introduced by Rignot et al. (2008) who scaled SMB anomaly with discharge. Box et al. (2013) isolated runoff as the discharge predictor after a sensitivity analysis. See Box et al. (2013) Figs. 1,2,3,4 and related text, including physical basis. Importantly to the concerns stated above, 1) Box et al. (2013) used 1958, 1964, and 1992-2009 data. Clearly, Fig. 4 in Box et al. (2013) show that 1958 and 1964 and other pre-acceleration years (1992-2004) lie near the regression lines. Furthermore, while 2000 through 2006 cover a changing period in Greenland discharge, there were likely other 'anomalous' periods in the past. There is independent evidence that the 1920s or 1940s may have been periods of significant discharge increase.**

**We have added some of this information to the manuscript.**

I have some concerns related to uncertainties as I could not find answers in the text. When the reconstructed SMB is scaled to the regional climate model data, is the dispersion added as an additional uncertainty to the scaled reconstruction? Similarly, when the ice discharge is modelled from the reconstructed runoff, is the dispersion added as an additional uncertainty?

**We now include the following text in the manuscript:**

**For reconstructed SMB and MMB, the mean of the recent uncertainty is added to the reconstructed uncertainty during the adjustment. Reconstructed MB uncertainty is then re-calculated as the square root of the sum of the squares of the reconstructed SMB and MMB uncertainty.**

**We implement that in the code, and the reconstructed time series uncertainty is visibly larger in Figure 2.**

Finally, it is assumed that the basal and surface mass balance uncertainties do not have time-invariant components, but I find this difficult to believe. What evidence is there for this?

**In the direct comparison between SMB observations and modelling, there is evidence that the errors are randomly distributed in time. Based on the limited data available which resolve SMB in time over longer periods, this seems to hold if you look at local comparison of ablation along the K-transect since 1990 (Fig. 9 in Noël et al. (2018)). Finally, comparison with time series from satellite products (similar epoch) does not show a clear deterioration/improvement in time (Fig. 7 in this paper). There is little information available before the 1990s to check for earlier epochs. In Overly et al. (2016), GPR derived accumulation support a stable uncertainty going back to 1980s.**

We have revised our uncertainty treatment and now assume MMB and BMB$_{GF}$ are time invariant (bias), but SMB, BMB$_{VHD}$, and BMB$_{friction}$ are time-varying (random).

Note however that these assumptions are only relevant for Fig. 4 where we accumulate errors in time. We do not use or provide this information for the output of this work - that is, the published data. There is only one error term there, and it includes both systematic and random error. We are not aware of other products that publish two error terms distinguishing between random and bias, but are eager to improve our methods if the editor or reviewers can provide a relevant citation with example.

We have added text to the "Accumulating uncertainties" section that clarifies this.

I would like to see each of the terms produced in this paper plotted and tabulated, i.e. SMB components (at least runoff and snowfall), ice discharge, submarine melting, basal melting.

The current work uses a single SMB term from the model outputs. At no point (in this work) do we separate SMB inputs and outputs. We clarify this in the text. Ice discharge and submarine melting are not (and I argue cannot at the moment) be separated. We have generated a Figure 2 with MMB and BMB separated, but do not believe it provides useful information, because BMB is so small relative to the other terms.

We are not sure how to present this in tabular form. There are too many days. Annual or decade? Average or sum? The data is accessible in an easy-to-access format (CSV) and NetCDF, and we provide code to display it.

2.2   Minor concerns

I have the following minor concerns:

L3: be clear about the period of data v models

Done.

L5: not sure you need to contrast performance to other mass balance estimates

Removed.

L6: and in any case which elements update daily?

Clarified. After 1986, all elements update daily (1 SMB term, all three BMB terms) except MMB, which updates every 12 days.

L9: define "general agreement"

Added range of r$^2$ values.

L10: say a little more about which other products are referred to here

Added.

L14: im not sure this statement is correct as there have been a large number of studies reporting trends in the ice sheet mass loss. We do know where, when, how, and why

the ice sheet has changed in mass. I think it would be a good idea to include a short summary here since it is absent elswehere in the introduction.

**We have re-written the 2nd sentence of the Introduction.**

L21: im not sure "no information" is correct. Locating the imbalance in space is informative

**Locating in space is certainly informative, but that is 'where' not 'how'. GRACE does not provide "information on the process contributing to changes in mass balance components (how)" (L19/20).**

L22: I don't think defintion is the correct word here, perhaps drop "definition of"

**Dropped.**

L24: integrated, not reduced

**We choose to keep 'reduced'.**

L24: suggest "typically" rather than "still"

**Changed.**

L26: i think you need to be careful about the sampling afforded by the satellites as compared to datasets made publicly available

**We are not sure exactly what you are referring to. VC products are usually higher spatial resolution than GMB products. We use the word 'satellite' throughout the document, but never with respect to spatial resolution. Here we are referring to public products.**

L26: this is not true; all methods give some information on processes its just integrated in the case of the raw VC and GMB methods but even in this case is informative because of the localisation. But in any case the methods are now routinely complemented also with RCM data to partition into dynamics and SMB as is done with IO. So I think the wording should be tightened up here. The IOM data are of clear value and its not neccessary to diminish the value of other methods

**We agree, but that 'integration' you refer to is precisely what removes the 'how'. Localisation is 'where', not 'how'. We are not sure what the 'complement' is, but believe RCM data is external to VC and GMB products, which is what we are discussing here. We have slightly re-written this paragraph to note that a) we do not have separate SMB terms as mentioned above and b) removed "is the only one".**

L32: Suggest "Inputs" not "IO inputs"

**Re-written, but we believe it is important to distinguish between "IO inputs" and "Inputs to this study", the latter which may include IO outputs (i.e., we use data from Mouginot et al. (2019)).**

L33: Suggest "Outputs" not "IO outputs"

**See above.**

L34: is it really annual for each component? I think you are oversampling the discharge for example and undersampling the RCM data

**Clarified that we are oversampling discharge. We are not undersampling RCM data - it is provided at daily resolution for the recent, and annual for the historical. RCMs may have O(minutes) timestep internally. We do not describe that here.**

L36: probably worth explaining to the reader what you mean here (presumably that the data will continue to be updated independent of this paper)

**Added 'daily-updating'.**

L37: suggest move the "Terminology" section to a glossary

**We prefer to keep it here.**

L38: "Product description" seems more like text for the acknowledgements or a footnote of some kind. In any case it should presumably come after the methods and results are reported

**We choose to keep it here - Because this is a 'data description' paper we believe it is appropriate to describe the output data near the beginning of the document. We will gladly re-arrange at the suggestion of the editor.**

L252: its only future relative to this dataset, its present relative to all the others

**Word 'future' removed.**

L280: suggest "start" instead of "top"

**Changed.**

L294: suggest add e.g. "relative to recent data" after "slope" to clarify

**Clarified.**

L307: do you mean data products not aligned, or their domains? Also, aligned in space (not time)

**We meant 'ice masks' and have changed the wording.**

L311: are you extrapolating data or interpolating it? I can't really tell. And are you resampling to a common grid?

**There is no interpolation. We are not sure how one would interpolate between 'CW' and 'NW' or sector 8.1 and 8.2 (without creating new sector 8.15). Therefore, extrapolation. The Zwally and Mouginot vector data is rasterized onto the native grid of HIRHAM, MAR, and RACMO.**

L350: what does large uncertainty mean?

**Removed 'large'.**

L413: not true, imbie data are equally weighting across all techniques present

**Adjusted text.**

[revised manuscript text omitted]

---

## Referee Report (RR1)

The concerns raised by reviewer 1 have been addressed in full. The major concerns raised by reviewer 2 (me) have been rebutted in most instances. I think more effort could have been made to reach a balanced response to both reviewers. I include some remarks below pertaining to the original concerns I raised that were rebutted. I leave it to the editor to take a decision on whether the authors response is satisfactory.

Marine mass balance

The new term "Marine Mass Balance" confused me in my first review. I was under the impression that it was defined to include submarine melting at the base of ice floating beyond a flux gate, but I realise now that it is in fact the sum of iceberg calving and melting at the calving front. I really think this is adding a layer of confusion to the literature and not clarifying things; ice discharge is a well-understood description and allows for the quantity to occur through either calving or melting at the calving front. In any case, the distinction is somewhat artificial because surface runoff (a component of the Surface Mass Balance) and subglacial runoff (a component of the basal mass balance) are both also discharged into the ocean, and could therefore be reasonably categorised as components of the marine mass balance. Moreover, there is in fact no purpose to the new definition as neither of the component terms (iceberg calving and calving front melting) are separated in this study. My advice is to avoid introducing confusion and revert to the established term "ice discharge", which is technically correct if one considers the inland side of the flux gate.

Modelled discharge

I raised a query relating to the period chosen to model ice discharge as a function of runoff. The authors have chosen 2000-2012, but this is an epoch when the ice sheet was in a considerable state of imbalance. I suggested checking how sensitive the results were to the choice of period; while the authors declined, I still believe this is a sensible thing to do.

SMB errors

I still find it difficult to believe that the SMB model output should have no time-invariant errors. Even though the authors dataset does not accumulate errors, in presenting an accumulated result in the accompanying paper the authors are implicitly recommended the approach.

Display of Mass Budget Terms

I suggested that the authors display each of the mass budget terms in the paper. The authors have replied that they are not sure how to do this – a response that lacks credibility. In the specific case of the BMB added to Fig. 2 perhaps the authors might consider a separate panel to avoid the problem of scale. In the case of tabulated results, perhaps the authors might consider annual values. The authors also state that at no point in this paper do they separate SMB into inputs and outputs, but in fact they do use runoff to model ice discharge which is what led me to believe that it was included in the study.